# Unraveling the hidden universe of small proteins in bacterial genomes

Samuel Miravet-Verde[1], Tony Ferrar[1], Guadalupe Espadas-García[2], Rocco Mazzolini[1], Anas Gharrab[1], Eduard Sabido[2], Luis Serrano[1,3,4,*] & Maria Lluch-Senar[1,3,**] iD

## Abstract

**Identification of small open reading frames (smORFs) encoding small proteins (≤ 100 amino acids; SEPs) is a challenge in the fields of genome annotation and protein discovery. Here, by combining a novel bioinformatics tool (RanSEPs) with "-omics" approaches, we were able to describe 109 bacterial small ORFomes. Predictions were first validated by performing an exhaustive search of SEPs present in *Mycoplasma pneumoniae* proteome via mass spectrometry, which illustrated the limitations of shotgun approaches. Then, RanSEPs predictions were validated and compared with other tools using proteomic datasets from different bacterial species and SEPs from the literature. We found that up to 16 ± 9% of proteins in an organism could be classified as SEPs. Integration of RanSEPs predictions with transcriptomics data showed that some annotated non-coding RNAs could in fact encode for SEPs. A functional study of SEPs highlighted an enrichment in the membrane, translation, metabolism, and nucleotide-binding categories. Additionally, 9.7% of the SEPs included a N-terminus predicted signal peptide. We envision RanSEPs as a tool to unmask the hidden universe of small bacterial proteins.**

**Keywords** mass spectroscopy; mycoplasmas; protein prediction; random forest classifier; small proteins

**Subject Categories** Microbiology, Virology & Host Pathogen Interaction; Post-translational Modifications, Proteolysis & Proteomics

**Mol Syst Biol. (2019) 15: e8290**

## Introduction

Development of ultra-sequencing technologies has led to a considerable increase in the number of annotated bacterial genomes (Kim *et al*, 2015). Classically, general genome annotation protocols only considered ORFs that encode for proteins larger than 100 amino acids (Angiuoli *et al*, 2008; Tatusova *et al*, 2016). This arbitrary cutoff was established to distinguish bona fide protein-coding ORFs from the numerous random in-frame arrangements of start and stop codons present in genomes (Crappé *et al*, 2013). However, recent studies have brought to light the importance of small open reading frame (smORF)-encoded proteins (SEPs; ≤ 100 amino acids; Ota *et al*, 2004; Savard *et al*, 2006; Makarewich & Olson, 2017), such as the antimicrobial peptides (AMPs) secreted by insects, animals, plants, and humans in response to infection (Avila, 2017).

In bacteria, SEPs exhibit a wide range of functions that are essential for the cell. SEPs can be involved in cell division (Blr, MciZ, and SidA), transport (AcrZ, KdpF, and SgrT), and signal transduction (MgrB and Sda) or even act as chaperones (FbpB, FbpC, and MntS; Storz *et al*, 2014). They are also involved in protein complexes, stress responses, virulence, and sporulation (Burkholder *et al*, 2001; Rowland *et al*, 2004; Alix & Blanc-Potard, 2008; Hemm *et al*, 2010; Lluch-Senar *et al*, 2015). Interestingly, these small proteins can also be used for communication between bacteria and phages, and as bacteriocins within niches like microbiota, thereby making them an important molecule to study when searching for new therapeutic protein candidates (Duval & Cossart, 2017).

Identifying SEPs is both technically and computationally challenging. At the experimental level, techniques such as ribosome profiling (Ribo-Seq; Mumtaz & Couso, 2015) and mass spectroscopy (MS; D'Lima *et al*, 2017) are typically used. However, as it is difficult to identify the translated frame in Ribo-Seq experiments, the identification of proteins encoded by overlapping ORFs is not feasible in most cases. Similarly, the absence of ribosome-binding sites (RBS, Shine–Dalgarno) in some bacterial genomes (Dandekar *et al*, 2000; Lluch-Senar *et al*, 2007), and the existence of mRNA without UTRs, makes it difficult to discern smORFs (Goyal *et al*, 2017). The detection of SEPs with common tryptic-based bottom-up MS proteomics approaches is also difficult due to the mere fact that their small size correlates with a reduced number of tryptic peptides (TPs; Yang *et al*, 2011; Saghatelian & Couso, 2015). Additionally, identification is further impeded by the fact that SEPs can be secreted, have relatively short half-lives, be present in low abundances, and exhibit tissue- and time-specific expression patterns (Goldberg, 1972; Wang *et al*, 2007).

1 EMBL/CRG Systems Biology Research Unit, Centre for Genomic Regulation (CRG), The Barcelona Institute of Science and Technology, Barcelona, Spain
2 Centre for Genomic Regulation (CRG), The Barcelona Institute of Science and Technology, Barcelona, Spain
3 Universitat Pompeu Fabra (UPF), Barcelona, Spain
4 Institució Catalana de Recerca i Estudis Avançats (ICREA), Barcelona, Spain
*Corresponding author. Tel: +34 93 3160101; E-mail: luis.serrano@crg.eu
**Corresponding author. Tel: +34 93 3160174; E-mail: maria.lluch@crg.es

Evolutionary pressure on genes leads to sequence conservation. As such, gene predictions by cross-species comparisons can be useful for predicting the existence of common proteins (Kimura, 1980; Ina, 1995; Makalowski & Boguski, 1998). However, in such sequence conservation analyses, the probability of overprediction becomes higher for shorter sequences (Ochman, 2002). Additionally, species-specific SEPs like the Sda protein of *Bacillus subtilis* (46 amino acids), which represses aberrant sporulation by inhibiting the activity of the KinA kinase, cannot be identified through comparative studies (Burkholder *et al*, 2001; Rowland *et al*, 2004). Furthermore, although computational methods based on the rate of synonymous and non-synonymous substitutions can differentiate between coding and non-coding regions, these alignment-based methods have two clear limitations. First, a closely related organism is required as a reference, and second, in order to avoid biases in the estimation, this type of method can only be applied to non-overlapping sequences (Lin *et al*, 2011). Other approaches are based on machine learning (ML) algorithms like interpolated Markov models (Salzberg *et al*, 1998), support vector machine-based classifiers (Kong *et al*, 2007), logistic regression (Kong *et al*, 2007; Zhao *et al*, 2016), and decompose–compose methods (Hu *et al*, 2016). These methods analyze the coding potential of a genome in an alignment-free manner without the need for experimental information. However, as these approaches do not take into account the importance of species-specific coding features in the classification, they prove inadequate for analyzing the genomes of organisms that are not considered in the training process itself. Importantly, none of these computational methods are free of biases when classifying overlapping annotations, a situation that is common for SEPs (Altschul, 1990; Mount, 2007).

Up until now, it has been difficult to determine the best method for comprehensively analyzing all putative SEPs. Here, by integrating more than 120 "-omics" datasets from *Mycoplasma pneumoniae*, we first assessed the experimental limitations of MS. Then, we developed RanSEPs, a random forest-based tool for the prediction of SEPs in any bacterial genome (Fig 1). We also validated the efficiency of RanSEPs by experimentally identifying SEPs in 12 bacterial species, including a set of 570 well-reported and experimentally characterized bacterial SEPs from different species (Hemm *et al*, 2010; Kodama *et al*, 2011; Storz *et al*, 2014; Baumgartner *et al*, 2016; Duval & Cossart, 2017; Impens *et al*, 2017; VanOrsdel *et al*, 2018). We also performed the same efficiency test on other protein discovery software and found that RanSEPs stands out as the best predictor. The higher prediction accuracy of our method is explained by the iterative randomization of the training set, a technique that enables the capturing of additional protein-related information during training. In addition, as the training sets are biased to include more SEPs, they place a higher level of importance on the possible alternative features of these proteins in the classification (Fig 1).

By applying RanSEPs to 109 bacterial genomes, we showed that the average number of SEPs per organism could be much higher than previously thought, with SEPs accounting for up to $16 \pm 9\%$ of the total coding ORFs. This result suggests that a remarkable number of bacterial SEPs remain unexplored, as recently reported (VanOrsdel *et al*, 2018). Additionally, even though most of the anti-sense non-coding RNAs (ncRNAs) are a product of transcriptional noise and dispensable for cell survival (Lluch-Senar *et al*, 2015; Lloréns-Rico *et al*, 2016), some of them could encode for proteins.

In fact, integration of RanSEPs predictions with transcriptomics data from 11 bacteria species revealed that a fraction of ncRNAs (1%, mostly antisense and intergenic) could encode for SEPs. Finally, functional analysis of SEPs revealed an enrichment in functions related to the membrane, translation, metabolism, and nucleotide binding. As previously described (Kemp & Cymer, 2014; Sheng *et al*, 2017), we observed a significant proportion of SEPs with N-terminus predicted signal peptide (9.7%) and transmembrane segments (15%). At a time when deep sequencing of microbiomes results in the identification of thousands of new bacterial species, our tool opens up the possibility to predict new SEPs that could modulate bacterial populations through quorum sensing or antimicrobial properties (Duval & Cossart, 2017).

# Results

## Key factors and criteria for the experimental identification of SEPs

To experimentally identify all SEPs encoded by the minimal genome of *M. pneumoniae*, we integrated both proteomics (116 MS experiments) and transcriptomics (eight experiments: four samples of RNA-Seq at 6 h, two at 24 h, and two at 48 h) experiments (Fig 1; Datasets EV1–EV3). Analysis of RNA-Seq and MS data was performed to identify possible new proteins having significant RNA expression and/or detected peptides. For this, we used a database including all putative proteins (length ≥ 10 amino acids) translated from the *M. pneumoniae* genome in all six frames (17,818 smORFs and 1,292 ORFs; see Materials and Methods; Fig 1). A "decoy" protein dataset of comparable size (Table 1), base composition and codon adaptation index (CAI) to that of *M. pneumoniae*, was used as a negative control to detect possible MS artifacts (Dataset EV3; see Materials and Methods).

Using MS, we identified 42 potentially new SEPs in *M. pneumoniae* with ≥ 1 unique tryptic peptide (UTP) and RNA expression levels ≥ 4.5 log2(counts) (Fig 2A; Datasets EV1 and EV3). However, 19 "decoy" SEPs were also detected (Fig 2B). While we found that the number of novel SEPs identified with ≥ 1 UTP increased in proportion to the number of experiments being considered, this same trend was also observed for the "decoy" SEPs (Dataset EV1 and Fig 2C). This trend suggested the existence of false positives in MS when considering no threshold for the number of identified UTPs. When we increased the number of detected UTPs to ≥ 2, we did not find any "decoy" protein but we did lose one NCBI-annotated SEP (Table 1 and Fig 2B) and the data quickly reached a plateau after four experiments (Fig 2C). The same happened using a threshold of one UTP and ≥ 1 non-unique tryptic peptide (NUTP). The number of putative SEPs was reduced from 42 to 7 using the first threshold and from 42 to 29 using the more relaxed threshold (Table 1, Fig 2B). After filtering by ≥ 2 UTPs, 532 proteins remained: 521 annotated, four novel standard proteins, and seven novel SEPs (shortest presenting a length of 48 amino acids).

To corroborate our ≥ 2 UTP threshold criteria, we performed targeted MS with C13C(6)15N(2)-labeled peptides of eight SEPs, four of which had ≥ 2 UTPs and four with one UTP (Dataset EV4). All four of the novel SEPs detected with ≥ 2 UTPs were confirmed with the C13 peptides. In contrast, we only detected a signal for two

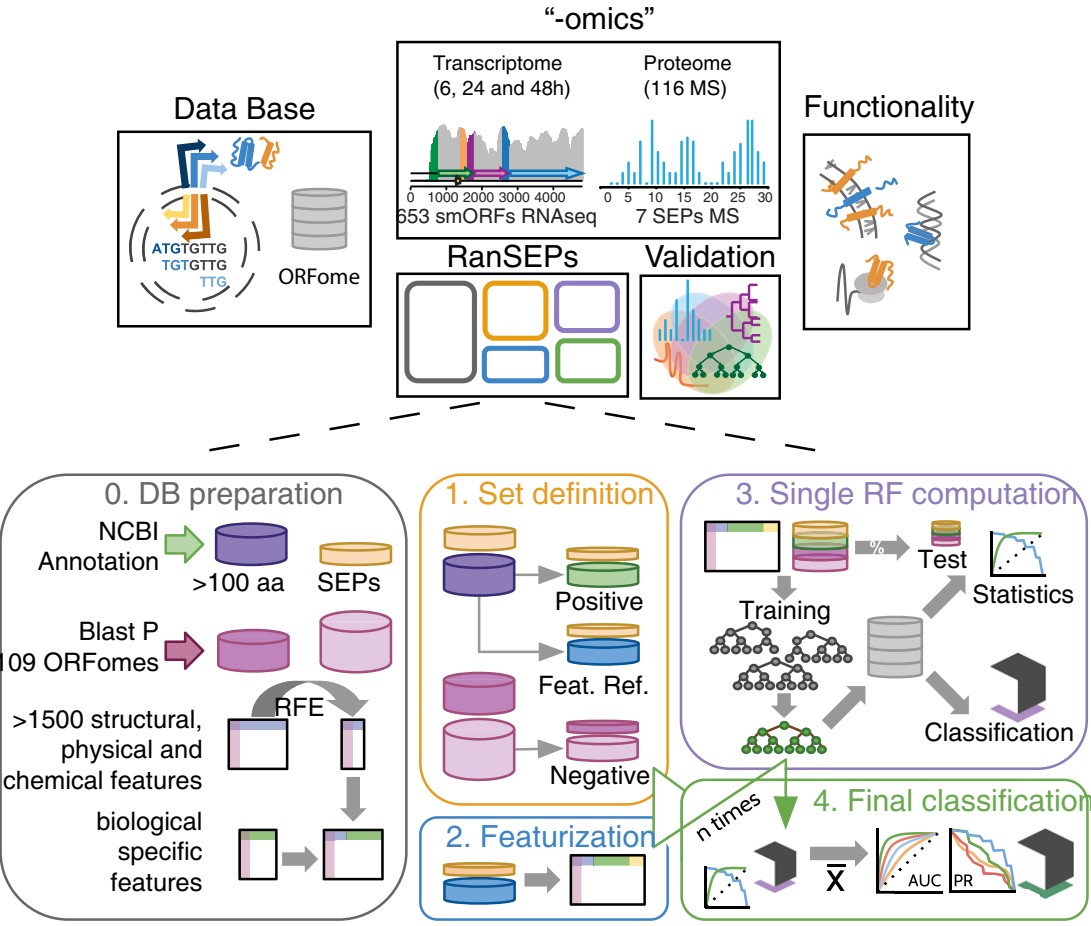

**Figure 1.  General workflow.**

First, we generated databases of all the putative ORFs encoded by the genomes of 109 different bacteria. The database of *M. pneumoniae* was used to perform the shotgun MS and RNA-Seq studies that were aimed at evaluating the coverage and performance of experimental approaches in the discovery of SEPs. In a parallel, experiment-independent manner, RanSEPs performed *in silico* predictions of potential novel proteins in the database. Results coming from both experimental and computational approaches are integrated in a validation step using a set of 570 SEPs characterized both in this work and in previous studies. Finally, RanSEPs predictions for the 109 bacterial genomes are combined together to assess the functional diversity and importance of predicted SEPs. The second part of the figure highlights how RanSEPs functions. In step 0 (gray box), RanSEPs detects annotated standard proteins (purple) and SEPs (yellow). By BLASTP, non-conserved standard and SEP proteins are detected (pink and light pink, respectively). In parallel, protein features are computed and filtered by Recursive Feature Elimination. These features are combined with general features of biological interest. In step 1 (yellow box), RanSEPs randomly subsets annotated standard and small proteins into a positive (green and yellow), a feature (blue and yellow), and a negative (pink and light pink) set from the bulk of non-conserved sequences. During step 2 (blue box), specific features that vary with each iteration are appended. In step 3 (purple box), the labeled positive and negative sets are divided into training and test sets. Step 4 (green box) consists of collecting the classifiers and classification task results, and computing the final statistics and scores for all the sequences. Step 0 is only run once, and then, it is out of the iteration process. Steps 1–3 are repeated as many times as iterations selected by the user. Step 4 is computed at the end to integrate the results of each iteration.

**Table 1.   Detection of SEPs using MS in *Mycoplasma pneumoniae*.**

| | Type | N | Criteria | | |
|---|---|---|---|---|---|
| | | | ≥ 1 UTP | ≥ 1 UTP; ≥ 1 NUTP | ≥ 2 UTP |
| SEPs | Annotated | 26 | 22 | 22 | 21 |
| | Putative | 17,792 | 42 | 29 | 7 |
| | Decoy | 20,100 | 19 | 0 | 0 |

The outcome of different results after MS searches using the decoy database (negative control) and translating all the possible ORFs in *M. pneumoniae*. When using the cutoff of at least 2 UTPs, the signal of every decoy protein was removed but the detection of putative SEPs consequently dropped, with one annotated SEP not being identified.

of the SEPs identified with one UTP in targeted proteomics (Appendix Figs S1 and S2; Accession number of MS results: PXD008243). These results indicate that ≥ 2 UTPs should be considered as the threshold for protein discovery without false positives, but that true SEPs could be lost.

Interestingly, 25% of the annotated proteins of *M. pneumoniae* were not identified by MS. By using PeptideSieve (Mallick *et al*, 2007), we measured the responsiveness of the proteins to MS. We found that annotated proteins detected by MS had a significantly higher number of high-responsive UTPs (HR_UTPs) than undetected proteins (Mann–Whitney one-sided *P*-value = 0.03; Dataset EV3 and Appendix Fig S3), revealing that not only the number of UTPs, but also their properties, could hamper protein detection by MS.

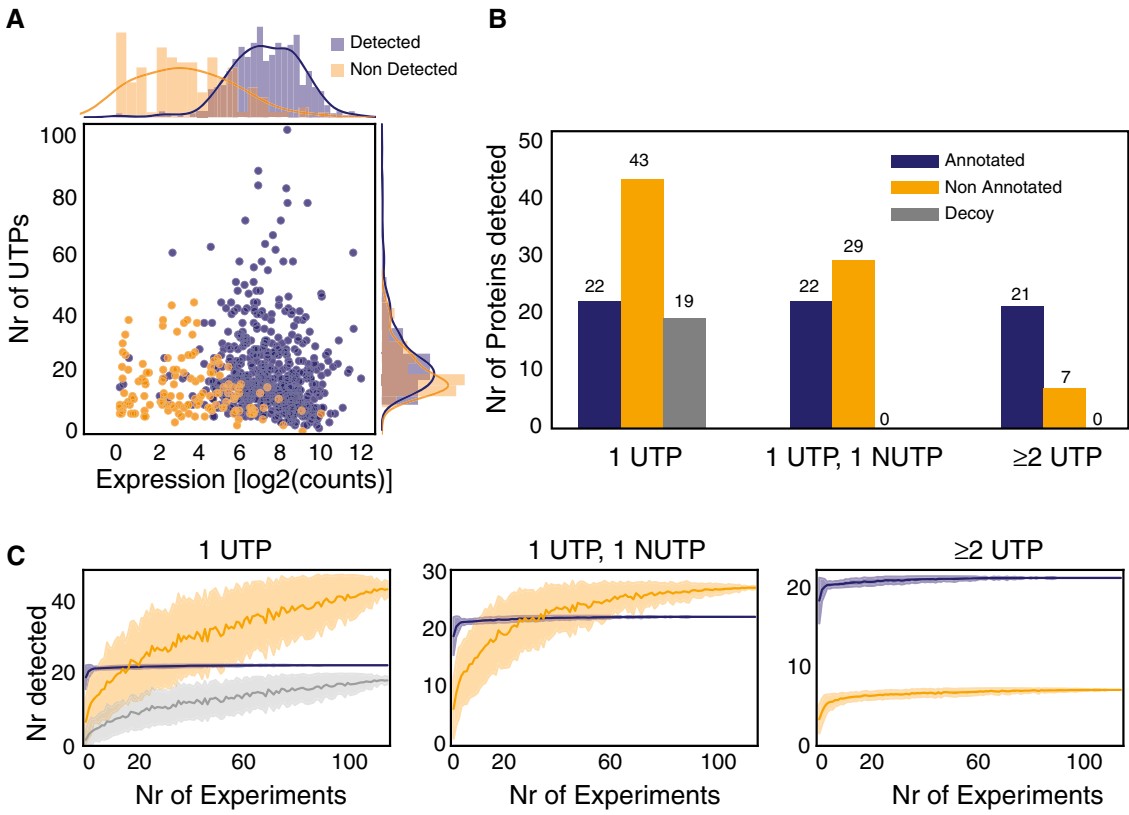

**Figure 2. Assessment of the detection coverage by "-omics" approaches.**

A  Evaluation of expression by RNA-Seq and number of peptides required to detect an annotated protein by MS in *M. pneumoniae*. The plot represents the relationship between expression levels (average expression from RNA-Seq data) and number of possible unique tryptic peptides (UTPs) for two sets of studied proteins: detected (blue dots) and not detected (orange dots) by MS.

B  Evaluation of thresholds and artefactual signals in MS data. The histogram represents the total number of SEP proteins detected in 116 shotgun MS experiments with 1 UTP, 1 UTP and 1 NUTP, or ≥ 2 UTPs for three categories. Color code: annotated (blue bars), putative new (orange bars), and decoy set (gray bars).

C  Number of SEPs detected by increasing the number of experiments. Color code is the same as in panel (B). Each line represents the accumulated number of different SEPs detected (*y*-axis) when combining 1–116 MS datasets (*x*-axis) from *M. pneumoniae*. Each line has an associated error that is shaded and represents the standard deviation within combinations of datasets (e.g., *x* = 80 will present the average number of proteins detected taking every combination of datasets in groups of 80 samples).

Analysis of the proteome (and its conservation) of five closely related Mycoplasmas revealed that 159 possible SEPs could be conserved in more than two species. Of these, we detected 48 by MS (Datasets EV5–EV10), 30 with 1 UTP, and 18 with at least 2 UTPs. While these 18 SEPs were identified with ≥ 2 UTPs in some of the species, in others, they were detected with only 1 UTP. This reinforces the idea that some SEPs having only 1 UTP could in fact be real. Therefore, conservation analysis could be helpful in identifying new SEPs as long as it is performed in conjunction with MS experiments in multiple organisms. Nonetheless, this approach could be misleading in the case of overlapping genes (Appendix Fig S4).

To confirm that the ≥ 2 UTPs criteria enable us to identify true proteins, we studied the correlation between ribosome profiling and the number of UTPs. For this purpose, we used raw datasets of ribosome profiling that were recently published for *Escherichia coli* (Hücker *et al*, 2017). We then analyzed an *E. coli* extract enriched in SEPs by MS (see Materials and Methods, Dataset EV11) and studied the correlation between both techniques. Ribosome profiling showed that the mRNA of SEPs detected with ≥ 2 UTPs presented significantly more bound ribosomes than both those detected with

just 1 UTP (Mann–Whitney one-sided test *P*-value = 0.005) and those not detected by MS at all (Mann–Whitney one-sided test *P*-value = 0.001, Appendix Fig S5). Thus, ribosome profiling supports using a ≥ 2 UTP cutoff to extract potential positive SEPs by MS.

In conclusion, while true-positive SEPs can be identified by MS when filtering by ≥ 2 UTPs, SEPs with only 1 UTP or very low responsiveness cannot be experimentally assessed by label-free proteomics. Therefore, experimental validation of SEPs still remains a challenge, and development of computational prediction tools capable of identifying SEPs without compromising the false discovery rate is paramount.

### RanSEPs: a novel random forest approach for the discovery of SEPs

Computational approaches are required not only to predict SEPs but also to reduce the required number of targeted validation experiments. For this purpose, we have developed RanSEPs, a variation of the random forest (RF) algorithm that iterates and randomizes training sets at the same time that it defines protein features (see

Materials and Methods). These features are selected in a blind manner by their importance in test classifications (Fig 1). With this approach, positive and negative set selections are fully randomized in each iteration, thereby generating an individual classifier each time. The positive sets comprise subsets of annotated proteins from NCBI that are forced to include a minimum percentage of SEPs belonging to the target organism. For the negative set, RanSEPs creates random sets of smORFs that are located within intergenic regions (relative to annotated genes) and have no identified homologs in a database including the six translated reading frames of 109 different organisms. Conceptually, this set could include actual SEPs; however, the probability of maintaining a true SEP and biasing the prediction is virtually null (see Materials and Methods).

The output is a probability score for a specific protein belonging to the coding class. When assigning the coding class to SEPs of *M. pneumoniae*, we set the threshold to a score $\geq 0.5$, while for standard proteins, it was set to a score $\geq 0.85$ ($95^{th}$ percentile for both distributions, Appendix Fig S6A). With the results of the previous prediction and using cross-validation, we obtained an average true-positive rate (TPR) of 96.3 and 90.3 % for annotated SEPs and standard proteins, respectively, with a total area under the ROC curve (AUC) of 0.92 when considering both types of proteins (Appendix Fig S6B and C). Using these settings, RanSEPs predicted 756 ORFs for *M. pneumoniae*: 612 standard proteins (598 annotated and 14 new) and 144 SEPs (26 annotated and 118 new). All of the new SEPs detected by MS with $\geq 2$ UTPs were classified by RanSEPs as coding (see Appendix Supplementary Methods and Figs S12–S14). Among the 23 SEPs detected with 1 UTP, RanSEPs predicted only five to be true, of which one was previously annotated with function while the other four were annotated by inference in closely related organisms. Interestingly, the other 18 putative smORFs with one UTP that were classified as non-coding by RanSEPs did not present homologous annotated candidates in closely related Mycoplasma species and their RNA levels were significantly lower compared with the five SEPs that had 1 UTP predicted as positive by RanSEPs (expression levels > $90^{th}$ percentile, Dataset EV3). This agrees with what we found in the previous section and supports the application of RanSEPs as tool for predicting those potential SEPs identified with 1 UTP by MS.

Next, we determined the proportion of predicted SEPs that could be considered false positives: pseudogenes and highly repeated sequences. Within the complete smORFome of *M. pneumoniae*, we detected 44 smORFs that could be derived from the fragmentation of a larger protein found in *M. genitalium* and 242 with at least two homologous matches in the *M. pneumoniae* genome. RanSEPs classified eight of the 242 "repeated" annotations as coding and predicted the 44 fragments to be non-coding (Dataset EV3). This homology information is integrated into every RanSEPs prediction to enable prioritization of results and provide more meaningful predictions.

### RanSEPs validation and method comparative

To validate RanSEPs predictions and test its potential applicability in other bacterial genomes, we generated a positive small protein set ($n = 570$) including multiple sources. First, MS was used to identify SEPs from enriched protein extracts of *Escherichia coli*,

*Pseudomonas aeruginosa*, and *Staphylococcus aureus* (see Materials and Methods, Datasets EV11–EV13), as well as from total protein extracts of six Mycoplasma species (Datasets EV6–EV10). Second, we re-analyzed publicly available MS datasets generated to detect SEPs and reported in the literature: *Lactococcus lactis* (PRD000266), *Synechocystis* sp. PCC6803 (PXD001246), and *Helicobacter pylori* (PXD000054; Datasets EV14-EV16; see Materials and Methods). In total, 473 SEPs (25 potentially new SEPs; 11 corroborated also by targeted proteomics) were found with $\geq 2$ UTPs in MS searches of these 12 bacterial species. Finally, 97 SEPs reported and validated in the literature were also added to this positive protein set (Dataset EV18). We also defined a balanced negative protein set ($n = 570$), which included 13 smORFs tested by targeted proteomics with negative results and 536 putative smORFs expected to be true negatives. This 536 smORFs subset was extracted from a collection of 14,746 putative smORFs from the 12 bacterial species studied by MS (Dataset EV18; see details in Materials and Methods). The criteria for selecting them were as follows: (i) They are not conserved in closely related species, and (ii) they have more than two high-responsive UTPs by Peptide-Sieve and are not detected by MS (Dataset EV18).

For validation, we performed specific RanSEPs predictions for each species, ensuring that the SEPs included in the validation set were never used in any training step (details about species-specific parameters can be found in Appendix Supplementary Methods). The same test was replicated with commonly used annotation prediction tools: CPC2, GeneMarkS, BASys, Glimmer, and Prodigal (Besemer *et al*, 2001; Van Domselaar *et al*, 2005; Delcher *et al*, 2007; Hyatt *et al*, 2010; Kang *et al*, 2017). One factor that makes RanSEPs different from other predictors is that it is able to compute and use species-specific feature weights to determine coding potential (see Materials and Methods). As specific features do not necessarily share the same general importance across different organisms, this functionality allows unbiased searches to be carried out for any organism. For example, the Shine–Dalgarno sequence, which acts as an RBS and has an important role in translation, is not always present in bacterial species, including Mycoplasmas (Fusaro *et al*, 2009; Omotajo *et al*, 2015). This can be observed when measuring the feature weights by RanSEPs, as RBSs, which are rarely found in *M. pneumoniae* genes (Weiner, 2000), have a very low weight in this organism (Fig 3A).

We assessed and compared the quality of the predictions in terms of accuracy and AUC (see Materials and Methods), and also in terms of computational cost (Appendix Fig S8, see Appendix Supplementary Methods). RanSEPs was the best tool for predicting SEPs (AUC = 0.95; accuracy = 0.89) as none of the other tools had an AUC > 0.85 (Dataset EV19, Fig 3B, and Appendix Fig S7). Remarkably, RanSEPs provided the best TPR (SEPs properly predicted as SEPs over total positives) for annotated proteins (86.8 %), SEPs with $\geq 2$ UTPs in MS (86.7 %), and potential new SEPs (76 %). It was also the only tool that predicted all the SEPs validated by targeted C13 proteomics without false positives (Dataset EV19, Appendix Fig S15). In terms of false-positive rates (smORFs wrongly predicted as SEPs over total negatives, FPR), RanSEPs returned the third lowest value, coming after BASys and CPC. However, these two tools did not reach TPRs higher than 65 %.

Finally, we further validated our prediction tool at the genome-wide level by studying the correlation between

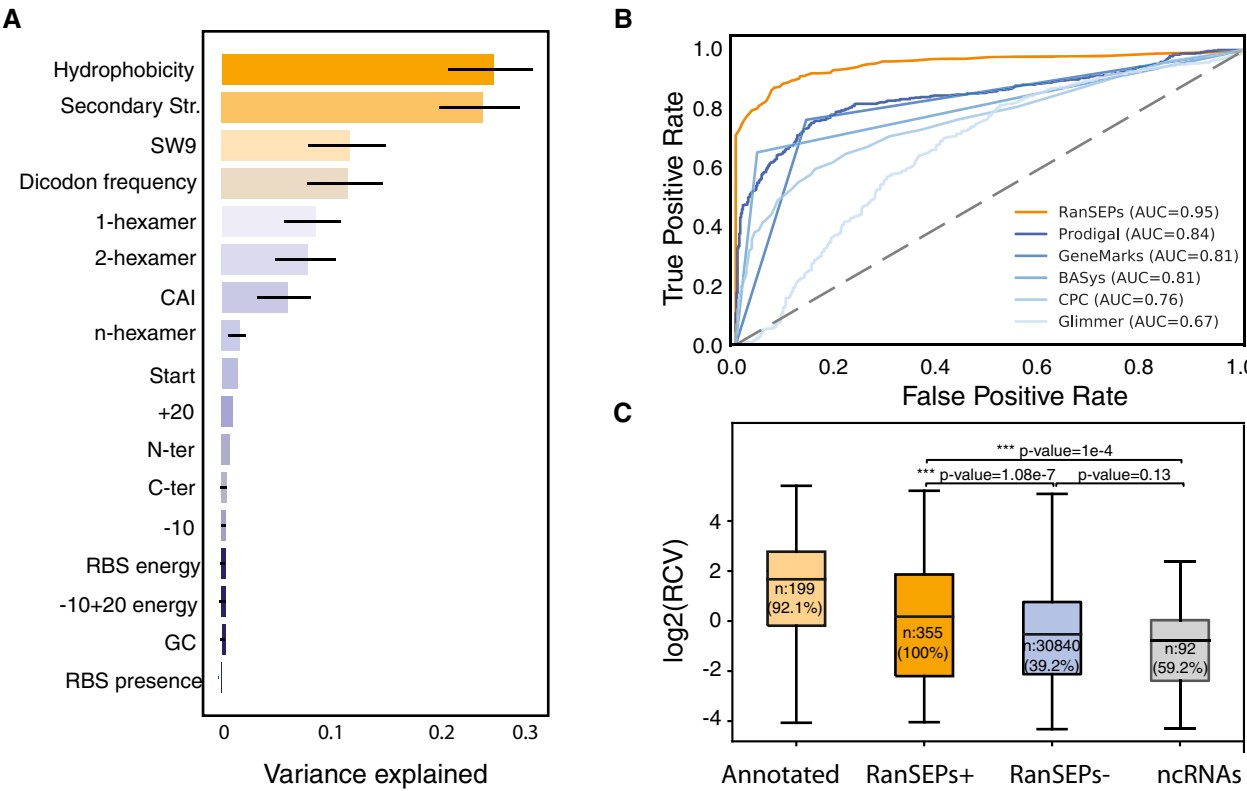

**Figure 3. RanSEPs predictions.**

A   Feature weight prediction in *M. pneumoniae*. Weights of the different features considered in the classification by RanSEPs. Bars indicate the global averaged variance that each feature explains by itself along with its associated standard deviation (black line) (25 iterations to estimate the error).

B   Method accuracy comparative. Receiver operating characteristic curve for RanSEPs (orange) and five additional tools (blue gradient). The closer a curve to the left-hand border, the more accurate the tool. The area under the curve (AUC) associated with each method is presented, with values closer to 1 indicating a more accurate method. The dashed gray line represents a classifier that assigns the coding class randomly.

C   Boxplot representing the relationship between RanSEPs-positive ("RanSEPs⁺", score ≥ 0.5) and RanSEPs-negative ("RanSEPs⁻", score < 0.5) SEPs predictions and associated RCV (ribosome profiling ratio coverage, in $\log_2$) in *Escherichia coli*. Only annotations ≤ 300 nucleotides in length were included. As positive and negative controls, we considered annotated SEPs ("Annotated") and non-coding RNAs ("ncRNAs"), respectively. Annotations within RanSEPs⁺, RanSEPs⁻, and ncRNAs overlapping with known annotated genes were excluded. Annotations with RCV = 0.0 are filtered out, and the number within the box represents the percentage of values in that class that are kept in the comparative. Along the top, *P*-values computed by Mann–Whitney rank test are indicated.

gene-expression-corrected Ribo-Seq coverage (RCV) and RanSEPs prediction in *E. coli* (Hücker *et al*, 2017). We found that SEPs predicted as positive showed significantly higher RCV levels compared with candidates predicted as negatives (Mann–Whitney one-sided test *P*-value = $1 \times 10^{-7}$) and ncRNAs (Mann–Whitney one-sided test *P*-value = $1 \times 10^{-4}$, Fig 3C). Additionally, while RanSEPs-positive predictions presented RCV values closer to the scores of annotated proteins, although still significantly lower (Mann–Whitney two-sided test, *P*-value = $1 \times 10^{-10}$), negative predictions were more similar to annotated ncRNAs (no significant differences by Mann–Whitney two-sided test, *P*-value = 0.13).

We next confirmed that the high success rate of RanSEPs was not due to an excess of positively scored annotations. In this analysis, we used the previously defined collection of 14,746 smORFs with low coding potential to search for false positives. Glimmer and CPC yielded the lowest FPRs but also had significantly limited TPRs. The rest of the tools presented comparable FPRs, with values of 5.1, 4.3, 3.6, and 3.9% for Prodigal, RanSEPs, BASys, and GeneMarkS, respectively. None of the false positives returned

by RanSEPs presented a score higher than 0.65, indicating that a stricter score threshold would prevent the detection of false positives. However, this threshold lead the average AUC falling to 0.88, indicating that we would miss valid SEPs. Additionally, RanSEPs provides extra information associated with the scores for further prioritization of the predictions. This information includes aspects like the presence of an RBS and a preliminary classification of the predicted SEPs into one of the following groups: (i) conserved in closely related species but not annotated in the organism of interest or any other; (ii) conserved and annotated in other species with a known function; (iii) conserved and annotated in other species without a known function; (iv) highly repeated in the annotated reference genome; or (v) potential pseudogene (see Materials and Methods).

### RanSEPs in a species-specific context and ncRNAs

To study the smORFomes in different bacterial genomes and to address the outstanding question regarding the percentage of coding

annotations represented by SEPs, we applied RanSEPs to 109 bacterial genomes. RanSEPs was parameterized and ran independently for each genome (see details in Materials and Methods and Appendix Supplementary Methods), and we considered the two thresholds defined above: the one that maximizes true positives (RanSEPs score ≥ 0.5) and the one that minimizes false positives in *M. pneumoniae* (score ≥ 0.65; Dataset EV20). This resulted in an average TPR of 86 ± 7% for annotated SEPs (iteratively excluding them from the training sets) with the 0.5 score, and 67 ± 12% with the 0.65 score. On average, the number of annotated SEPs over the total number of annotated coding ORFs was 10 ± 5%, a value that reaches 16 ± 9.5% when adding SEPs predicted by RanSEPs with a score of ≥ 0.5 and 14 ± 7% when raising it to ≥ 0.65 (Dataset EV20). On average, we determined that 1 ± 0.7% of the SEPs predicted by RanSEPs with a score ≥ 0.5 could be considered pseudogenes or "repeated" sequences when the SEP was a fragment of a larger protein in another organism or found several times in the reference genome. These values were reduced to 0.75 ± 0.1% when using the ≥ 0.65 threshold. Ultimately, this implies that between a minimum of 13 ± 7% and a maximum of 16 ± 9.5% of the proteins in each genome could be SEPs (Dataset EV20). The prediction results for the 109 bacteria can be downloaded at www.ranseps.crg.es.

As in *M. pneumoniae*, secondary structure and hydrophobicity were the most important features for polypeptide classification in all bacteria (Fig 4). However, some features like the SW9 and the four dicodon frequencies (see Materials and Methods) showed weight differences that resulted in two clusters of bacterial species. The first cluster (higher weight for SW9 and lower values for dicodon frequencies) presented higher rates of encoded SEPs than the second cluster (low weight for SW9 and high weight for dicodon frequencies, unpaired *t*-test *P*-value = 0.04). In addition, we observed that organisms with higher percentages of SEPs (> 13.16%, N = 55) were associated with bacteria having low GC contents (38 ± 12%). In contrast, lower rates of SEPs (≤ 13.16%, N = 54) were predicted for bacterial species with higher GC contents (47 ± 10%, unpaired *t*-test *P*-value = 0.005, Dataset EV20). These results agreed with previous studies, suggesting that a low GC content increases the number of stop codons and consequently results in an increased percentage of SEPs (Oliver & Marín, 1996; Mir *et al*, 2012) (see Appendix Supplementary Methods).

Over the past few years, it has been shown that sequences formerly described as ncRNAs could, in some cases, actually encode for proteins, with some of them being SEPs (Friedman *et al*, 2017; Hücker *et al*, 2017). Thus, we combined RanSEPs predictions with the annotated ncRNAs of 11 bacterial transcriptomes (Lloréns-Rico *et al*, 2016) and found that 273 out of 8,056 ncRNAs could in fact encode for 289 proteins: 184 SEPs and 105 standard proteins (Dataset EV21). Out of these 273 ncRNAs that could encode for

proteins, 11 (4%) were overlapping in sense with genes, 185 (67.8%) in antisense, and 77 were located in intergenic regions (28.2%; Appendix Fig S9). The average length of the 184 SEPs encoded by these re-annotated RNAs is 96 amino acids. In contrast, standard proteins encoded by former ncRNAs had an average length of 132 amino acids.

### Functional assessment of novel SEPs

In total, 36,311 SEPs were collected, including annotated and predicted SEPs from the 109 genomes considered. Out of this group, while 25,229 SEPs were found annotated in their original genomes (231 ± 186 annotated SEPs per genome), the majority of them were annotated as hypothetical proteins or with unknown function. In fact, only 5,175 SEPs (20%) were associated with a function. The majority of the SEPs with assigned functions were involved in translation (mainly ribosomal proteins), metabolism, and DNA/RNA binding (Fig 5A; Dataset EV22).

The total number of predicted SEPs not previously considered in their respective original reference genome was 14,773 using the ≥ 0.5 score criteria. To explore the possible functions of the proteins belonging to this group, we ran a BLAST search using the first group of SEPs with annotated functions as a database. Results indicated that, on average, a specific SEP with an undescribed function could be conserved in at least 15 different organisms (Appendix Fig S10). In addition, this analysis revealed that while 3,535 SEPs (24%) did not have annotated homologs, 11,238 (76%) were found annotated in other species: 5,038 (34%) with unknown function and 6,341 (42%) with different functions (Fig 5B). We repeated this search with the "decoy" protein dataset used for MS as the target, and found that no sequence passed the thresholds required to be considered homologous. As such, we would not expect to have false positives by chance. Although we have assigned functionality to most of the predicted SEPs in the 109 genomes, one needs to be cautious as sequence homology and functional annotation of small proteins is not always reliable.

Finally, in some bacteria, SEPs are known to be secreted and can play a role in communication or even act as toxins (Duval & Cossart, 2017). To determine whether some of the new SEPs we discovered could be secreted or be integrated into the membrane, we searched for signal peptide sequences as well as for transmembrane regions using Phobius (Käll *et al*, 2004). We focused on the set of SEPs with unassigned function and found that 9.7% had a N-terminus predicted signal peptide sequence and 15% a transmembrane membrane region (Dataset EV22). The percentage of SEPs with a signal peptide was higher than expected by chance when compared with the same "decoy" set of SEPs used in MS (9.7% for predicted SEPs, 1.2% for "decoy" SEPs, unpaired two-tailed *t*-test *P*-value = 0.018). Moreover, to confirm that the results obtained with

**Figure 4.  A comparison of the feature weights used for the prediction of SEPs in 109 bacterial genomes.**

Clustered heat map using nearest point algorithm and representing the weights of different features in 109 bacterial genomes, and the clustering relations between features (top dendrogram) and species (side dendrogram). Rightmost light-orange and light-blue bars are included to differentiate the two main clusters. Numbers in the right vertical axis are short references representing the names of the bacterial genomes (Dataset EV15). The right three columns represent biological features not used in the classification. The ratio of the percentage of SEPs compared to the median value is colored as blue and orange for ≤ 13.16% of SEPs and > 13.16%, respectively. Blue and orange colors in the % GC column represent genomes with ≤ 38 and > 38% GC content (median value = 38), respectively. Genome size column separates species into small-genome bacteria (≤ 1.5 Mb, blue) and large-genome bacteria (> 1.5 Mb, orange).

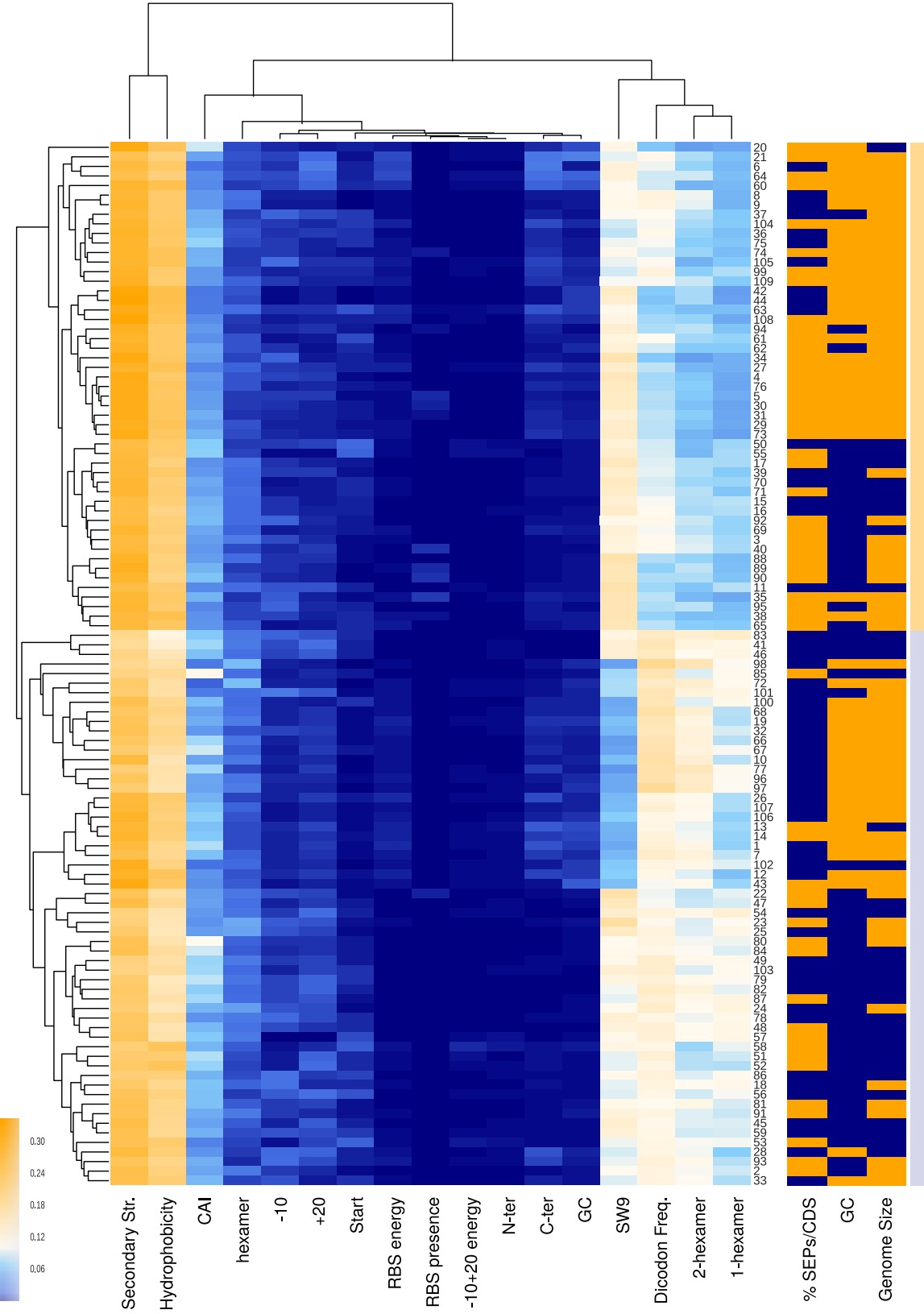

**Figure 4.**

Phobius were meaningful with regard to SEPs, and that protein size did not bias the analysis, we ran a test on a set of annotated standard proteins in which we sequentially shortened their C-terminus. The sensitivity of Phobius is higher than 80% for sequences over 30 amino acids. For sequences under 30 amino acids, however, we see values lower than 50%; this is expected when considering that Phobius specifically searches for a motif presented by the first 16–30 amino acids of the N-terminus of a protein. If the motif is located within these first amino acids and is short, Phobius will still detect the protein as positive (see Appendix Supplementary Methods and Fig S11).

## Discussion

Genome annotations, which traditionally considered only standard proteins, ignored the existence of a layer of complexity represented by SEPs (i.e., the smORFome). After assessing the experimental limitations, we showed that the experimental detection and characterization of SEPs are challenging. On the one hand, as "decoy" proteins sequences are detected by MS, proteins that do not exist can actually have spectra assigned (1 UTP or 1 UTP;1 NUTP). On the other hand, as these "decoy" proteins appeared across multiple experiments, discrimination criteria based on reproducibility are not

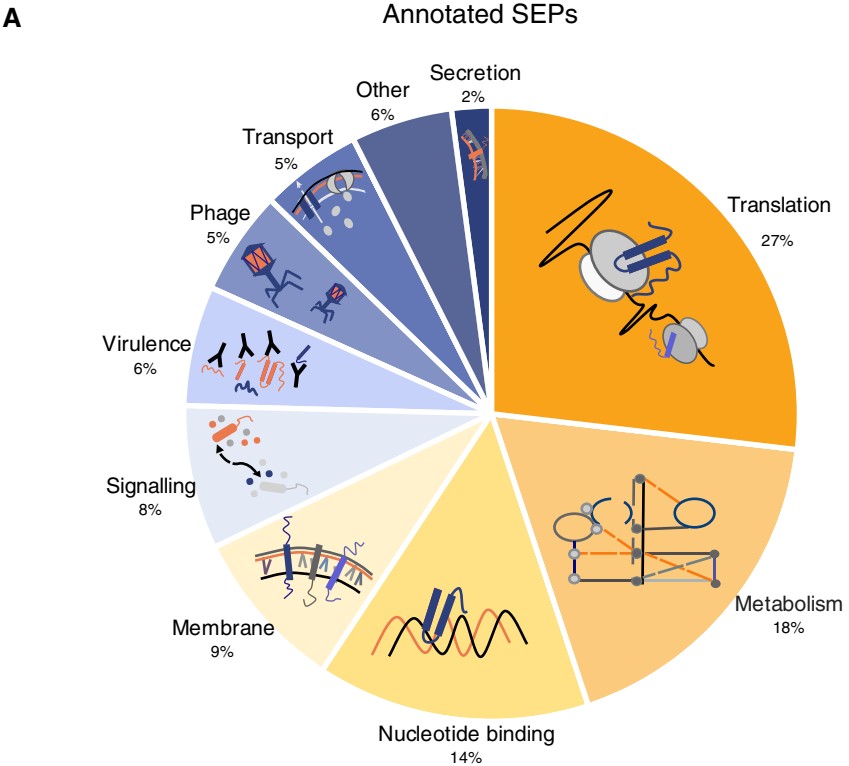

**A**   Annotated SEPs

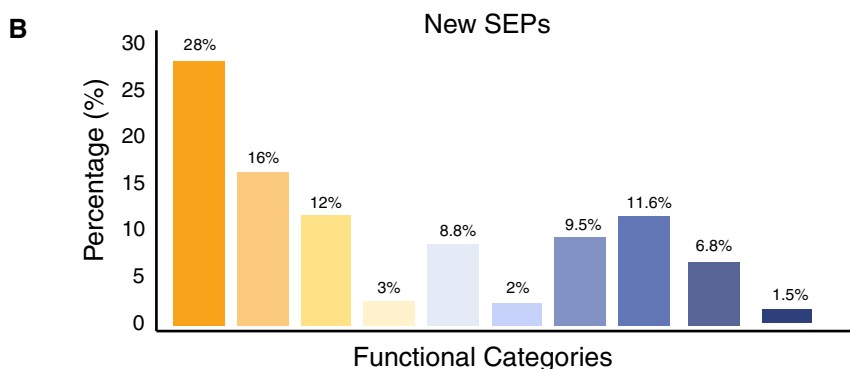

**B**   New SEPs

**Figure 5. Functional assessment of RanSEPs results.**

A   Landscape of the SEPs with functional annotations in NCBI considering 109 bacterial genomes (Number of SEPs = 25,229 SEPs).
B   Functional inference of the predicted SEPs (N = 11,238) as determined using BLASTP against NCBI-annotated SEPs having an associated function (N = 5,175). The color code associated with each category is the same as in panel (A).

feasible. This problem is solved by only accepting proteins detected with $\geq 2$ UTPs. These criteria were corroborated by re-analyzing Ribo-Seq data from *E. coli*. The main drawback is that many SEPs have very few responsive UTPs and consequently they are discarded. Despite these constraints, however, we were still able to detect novel SEPs in *M. pneumoniae* by integrating 116 shotgun MS datasets. Thus, this represents the first comprehensive study of a bacterial proteome using MS without protein size thresholds. Label-free MS experiments on cell extracts and SDS gel extraction derived from 12 bacterial species identified 25 new SEPs not annotated in the reference genomes. Of course, the problem associated with only 1 UTP could be partly alleviated by doing targeted proteomics with labeled C13 peptides. However, taking into account the required number of experiments and the fact that many SEPs do not have high-responsive peptides, the extensive analysis of SEPs encoded by a bacterial genome would be precluded. In addition, other factors could contribute to this problem like short protein half-lives, conditional gene expression, or special features in sequence associated with concrete functions (e.g., hydrophobicity).

Here, we developed RanSEPs to address the aforementioned limitations. Using *M. pneumoniae* as a reference, we developed RanSEPs as a predictor to define candidates given a specific genome and to score them by assigning a probability of being coding smORFs. Also, the assigned score provides meaningful information about features that can be important for the functional characterization of SEPs. Furthermore, we validated this application in other bacteria with SEPs that had been experimentally identified or described in the literature. Comparison of RanSEPs with five other tools showed that RanSEPs maximizes the correct prediction of true positives without increasing the false-positive rate. This could be attributed to its iterative method in which multiple classifications are averaged using different sets of annotated proteins in each iteration. This property permits the capture of a wide diversity of features presented by annotated genes, thereby resulting in more accurate predictions. Derived from this and considering that closely related species share sequence features, our scoring algorithm could also be modified to *de novo* annotate a genome of interest. In addition, the relationship between gene-expression-corrected ribosome profiling in *E. coli* and RanSEPs predictions showed that predicted SEPs generally have higher ratios than those predicted to be negative and resembling annotated ncRNAs.

Analysis of features that discriminate coding sequences in 109 bacterial genomes revealed that hydrophobicity and secondary structure are key factors. Also, we observed that the number of predicted SEPs encoded by a genome depends on the GC content. On the other hand, importance of features governing coding potential is conserved across species. Strikingly, between a $13 \pm 7$ and $16 \pm 9.5\%$ of the genes (depending on the cutoff score used) in these 109 species encoded for SEPs, highlighting that the coding capacity of bacterial genomes has likely been underestimated. Noteworthy, genome annotations are critical for classifying a SEP as a new protein. In fact, for 76% of the SEPs predicted by RanSEPs, orthologous SEPs were identified by BLAST in closely related strains. This result indicates that reference genomes are still incomplete and not properly curated.

Possibly, some of the predicted SEPs could be pseudogenes or false positives. Identification via homology of mutations resulting in a premature stop codon can provide an estimation of the number of pseudogenes present in a genome of interest. With this approach, we estimated that $1 \pm 0.7\%$ of predicted SEPs could be pseudogenes. These genes can be excluded, however, by increasing the RanSEPs threshold, albeit at the cost of missing some true SEPs. Thus, our 13–16% lower and upper estimates could still contain false positives but still represent a significant percentage.

Interestingly, some ncRNAs of multiple bacterial species could actually encode for proteins. While 63% of the proteins potentially encoded by ncRNAs were SEPs, 37% were standard proteins with an average amino acid length of 132 amino acids. This suggested that some ncRNAs could in fact be coding and that bacterial annotations could be missing not only SEPs but also longer proteins.

Functional analysis of the predicted and previously identified SEPs indicated that these proteins participate in basic processes of living systems such as transcription, translation, metabolism, signaling, quorum sensing, virulence, and pathogenicity. However, this analysis should be taken with caution as sequence homology and functional annotation of SEPs is challenging (VanOrsdel *et al*, 2018). Interestingly, similar to what has been previously reported (Kemp & Cymer, 2014; Sheng *et al*, 2017), we found a significant enrichment in SEPs presenting features indicative of being secreted (10%) or membrane localized (15%). This observation could have an impact not only on translational research but also on the study of the modulation of bacterial populations in microbiomes, thereby opening up a new line of research in the Systems Biology discipline (Duval & Cossart, 2017).

With all our results in mind, we envision RanSEPs as a tool to help predict new SEPs, support detections, and discard artifactual proteins detected by MS that have low signals such as those detected with only one UTP and/or one NUTP. When no experimental information is available, RanSEPs can help guide the selection of potential new SEPs for validation and further characterization with the overall aim of uncovering their functions.

## Materials and Methods

### ORFome database generation

We generated *in silico* proteomes by translating all putative ORFs with sizes $\geq 10$ amino acids from the six possible open reading frames of 109 bacteria. These bacteria included representative species of both gram types, and covered a wide spectrum of genome sizes (0.5–9 Mb), GC contents (20–70%), and generation times (0.48–12 h). Putative ORF databases were computed considering the codon translation table 11 (start codons: ATG, GTG, and TTG; stop codons: TAG, TAA, and TGA) for all cases except Mollicutes, which were based on translation table 4 (start codons: ATG, GTG, and TTG; stop codons: TAG and TAA). In all cases, only ORFs encoding theoretical proteins of at least 10 amino acids were accepted in the databases (www.ranseps.crg.es).

### Decoy database generation

A "decoy" dataset to assess the presence of possible artifacts when searching SEPs in a specific organism was generated based on certain factors. First, we used a comparable number of SEPs and standard proteins to the number in the target organism as it is

known that the database size can bias MS searches. Second, we forced the sequences to present a GC content and codon usage similar to those of the target organism. Lastly, we permitted only sequences that were not found in other organisms (BLASTP e-value > 0.1). In the end, the "decoy" dataset was composed of: (i) 2,433 translated stop-to-stop non-coding regions of *M. pneumoniae* without any start codon, ("in" prefix); (ii) 1,425 translated intergenic regions from the *M. pneumoniae* genome (without start codon, no overlap with any putative ORFs, "or" prefix); (iii) 8,740 pseudo-randomly generated peptides with a codon usage and GC content comparable to that of the *M. pneumoniae* genome, lengths between 20 and 100 amino acids, forced to have an average of three detectable UTPs, and comparable start and stop frequencies for start (ATG = 0.86, GTG = 0.073, TTG = 0.067) and stop: (TAA = 0.71, TAG = 0.28) codons (prefix "gc"); and (iv) 9,110 amino acid sequences obtained by translating the *in silico* random genome, preserving the GC content and codon usage of the *M. pneumoniae* genome (prefix "rd"). This genome is generated using frequencies and sizes of intergenic and coding regions similar to those of the annotated genome in NCBI. As GC content varies between coding and intergenic regions, we adjusted the "decoy" gene regions by codon adaptation index (CAI) and the intergenic ones by GC.

## Bacterial strains and growth conditions

*Mycoplasma pneumoniae* M129 was grown in 75-cm$^2$ tissue culture flasks with 50 ml of modified Hayflick medium at 37°C as previously described (Yus *et al*, 2012). *M. genitalium* G-37 (wild-type) strain was grown in SP-4 medium (Tully *et al*, 1979) at 37°C under 5% $CO_2$ in tissue culture flasks (TPP). *M. gallisepticum* str. R (high), *M. hyopneumoniae* 232, *M. capricolum* subsp. capricolum ATCC 27343, and *M. mycoides* subsp. capri str. GM12 were all grown as suspension cultures in SP-4 medium at 37°C and 200 rpm. *E. coli*, *S. aureus*, and *P. aeruginosa* (strain PAO1) were grown overnight in 22 ml TSB medium, at 37°C, shaking at 180 rpm.

## RNA extraction and library preparation for RNA-Seq

After growing *M. pneumoniae* for 6 h at 37°C, cells were washed twice with PBS and lysed with 700 μl of QIAzol buffer. RNA extractions were performed using the miRNeasy Mini Kit (Qiagen) following the instructions of the manufacturer. Libraries for RNA-Seq were prepared following directional RNA-Seq library preparation and sequencing. Briefly, 1 μg of total RNA was fragmented into ~100–150 nt using NEB Next Magnesium RNA Fragmentation Module (ref. E6150S, NEB). Treatments with Antarctic phosphatase (ref. M0289S, NEB) and PNK (ref. M0201S, NEB) were performed in order to make the 5′ and 3′ ends of the RNA available for adapter ligation. Samples were further processed using the TruSeq Small RNA Sample Prep Kit (ref. RS-200-0012, Illumina) according to the manufacturer's protocol. In summary, 3′ adapters and subsequently 5′ adapters were ligated to the RNA. cDNA was synthesized using reverse transcriptase (SuperScript II, ref. 18064-014, Invitrogen) and a specific primer (RNA RT Primer) complementary to the 3′ RNA adapter. cDNA was further amplified by PCR using indexed adapters supplied in the kit. Finally, size selection of the libraries was performed using 6% Novex® TBE Gels (ref. EC6265BOX, Life Technologies). Fragments with insert sizes

of 100–130 bp were cut from the gel, and cDNA was precipitated and eluted in 10 μl of elution buffer. Double-stranded templates were cluster-amplified and sequenced on an Illumina HiSeq 2000. The raw data of RNA-Seq were submitted to the ArrayExpress database (http://www.ebi.ac.uk/arrayexpress) and assigned the identifier: E-MTAB-6203.

For each experiment, both ends were treated as independent single-end reads in order to avoid the wrong assignment of read-pairs. Filtered reads were mapped to each reference genome using Maq mapping software. We mapped the reads containing 50 bp, allowing for one mismatch. The expression per ORF was computed based on:

$$\text{Expression} = \log 2(\text{counts per gene/gene length}).$$

To define an ORF as "transcriptionally active", its expression value had to pass a threshold established by the minimum expression value for all previously annotated genes of the organism of interest.

## Prediction of possible and high-responsive UTPs

To determine the number of expected high-responsive UTPs, we used PeptideSieve (Mallick *et al*, 2007) with the default properties file and selecting results for "Page Electrospray: PAGE_ESI" with a probability score > 0.65. This threshold was selected as it provided the best correlation between predicted UTPs and those observed experimentally (0.61 correlation coefficient). A peptide was considered to be a UTP only when it was found to be associated with one protein and have a minimum length of 5 amino acids (Dataset EV3).

## Mass spectrometric analyses

### Sample preparation

To generate new samples for MS analysis, 5 ml of the *P. aeruginosa*, *E. coli*, and *S. aureus* overnight cultures was centrifuged and resuspended in 500 μl lysis buffer (20 mM sodium phosphate, pH 7.4, 500 mM NaCl, 1% Triton, 2 mM DTT + protease inhibitors + lysozyme 50 μg/ml). Then, the lysates were incubated 20 min at RT, disrupted by sonication (15 min × hi 30'' on/off on ice), and centrifuged for 30 min at 21,130 *g*. Twenty microliters of both the supernatant and the pellet was loaded on Novex 10–20% Tricine gels (Thermo Fisher # EC6625BOX) and run at 120 V for 30 min. Afterward, different portions of the gel were cut with a scalpel: one portion below the loading buffer line, and the other portion between the loading buffer line and the 10 kDa marker.

Data from 116 shotgun MS experiments corresponding to different mutants and conditions of *M. pneumoniae,* as shown in Datasets EV1–EV3, were re-analyzed with the new database (see above) to re-annotate the *M. pneumoniae* genome (ID PRIDE: PXD008243).

Samples extracted with SDS were reduced with dithiothreitol (90 nmols, 30 min, 56°C), alkylated in the dark with iodoacetamide (180 nmols, 30 min, 25°C), and digested first with 3 μg LysC (Wako, cat # 129-02541) overnight at 37°C and then with 3 μg of trypsin (Promega, cat # V5113) for 8 h at 37°C following the fasp produce of Wiśniewski (2016). Samples extracted with urea were reduced with dithiothreitol (90 nmols, 1 h, 37°C) and alkylated in

the dark with iodoacetamide (180 nmol, 30 min, 25°C). The resulting protein extract was first diluted 1/3 with 200 mM NH$_4$HCO$_3$ and digested with 3 μg LysC (Wako, cat # 129-02541) overnight at 37°C, and then diluted 1/2 and digested with 3 μg of trypsin (Promega, cat # V5113) for 8 h at 37°C.

After digestion, the peptide mix was acidified with formic acid and then desalted with a MicroSpin C18 column (The Nest Group, Inc) prior to LC-MS/MS analysis.

*Sample acquisition*
The peptide mixes were analyzed using a LTQ-Orbitrap Velos Pro mass spectrometer (Thermo Fisher Scientific, San Jose, CA, USA) coupled to an EasyLC [Thermo Fisher Scientific (Proxeon), Odense, Denmark]. Peptides were loaded onto the 2-cm Nano Trap column, which had an inner diameter of 100 μm and was packed with C18 particles of 5 μm (Thermo Fisher Scientific), and were separated by reversed-phase chromatography using a 25-cm column that had an inner diameter of 75 μm and was packed with 1.9-μm C18 particles (Nikkyo Technos Co., Ltd. Japan). Chromatographic gradients were started at 93% buffer A and 7% buffer B with a flow rate of 250 nl/min for 5 min and were then gradually increased to 65% buffer A and 35% buffer B over 60 or 120 min depending on the complexity of the sample. After each analysis, the column was washed for 15 min with 10% buffer A and 90% buffer B (buffer A: 0.1% formic acid in water; buffer B: 0.1% formic acid in acetonitrile).

The mass spectrometer was operated in DDA mode, and full MS scans with 1 micro scans at a resolution of 60,000 were used over a mass range of m/z 350–2,000 with detection in the Orbitrap. Auto gain control (AGC) was set to 1E6, dynamic exclusion to 60 s, and charge state filtering disqualifying singly charged peptides was activated. Following each survey scan of each cycle of the DDA analysis, the top twenty most intense ions with multiple charged ions above a threshold ion count of 5,000 were selected for fragmentation at a normalized collision energy of 35%. Fragment ion spectra produced via collision-induced dissociation (CID) were acquired in the Ion Trap, with an AGC of 5e4, an isolation window of 2.0 m/z, an activation time of 0.1 ms, and a maximum injection time of 100 ms. All data were acquired using Xcalibur software v2.2.

*Database search*
Proteome Discoverer software suite (v2.0, Thermo Fisher Scientific) and the Mascot search engine (v2.5, Matrix Science) were used for peptide identification and quantification (Perkins *et al*, 1999). Samples were searched against a customized database for each species as described in the corresponding section. Trypsin was chosen as the enzyme, and a maximum of three miscleavages were allowed. Carbamidomethylation (C) was set as a fixed modification, whereas oxidation (M) and acetylation (N-terminal) were used as variable modifications. Searches were performed using a mass accuracy enforcement of 7 ppm, which goes accordingly with the accuracy of the Orbitrap mass analyzer, and a product ion tolerance of 0.5 Da. Resulting data files were filtered for FDR < 1.

*Targeted MS*
MS1 Targeted Area Extraction was performed with Skyline v3.7.011317 and using RAW files acquired in the Orbitrap Velos Pro that contained heavy-labeled internal standards (Dataset EV4).

## Conservation analyses: detecting homology and potential pseudogenes

An ORF was considered as conserved when it was found in three or more species. Three different thresholds were taken into account to assess the presence of the annotation in different bacteria. These thresholds were applied to the results by running a BLASTP of the amino acid sequence of the ORF of interest against a protein database comprised of a complete six-frame genome translation of 109 different bacterial species. Filter parameters included the e-value, the percentage of target sequence aligned, and the difference in length between the target and the hit. Thresholds for the three parameters were computed using the annotated proteins of the organism of interest as a reference. In the case of *M. pneumoniae,* 95% of the annotated proteins (with no size discrimination) have e-values smaller than $3 \times 10^{-8}$, more than 75% of their lengths aligned, and differ with the matched hit in < 20% of their length. We considered closely related species those sharing > 75% of their annotated proteins when applying the previously explained parameters.

Taking advantage of the conservation study, we implemented in RanSEPs an additional classification task to detect potential pseudogenes or highly repeated annotations that could be artifactually considered as coding. With this in mind, we classified every ORF into seven groups: 0—no hits passed the thresholds defined; 1—conserved with an annotated function; 2—conserved as an annotated SEP in NCBI but no associated function; 3—conserved in a different species but target and homologous sequence not found in NCBI; 4—sequence is completely or partially (> 75%) repeated ≥ 3 times in the reference genome; 5—potential pseudogene; and 6—to depict those annotations that are found in the reference NCBI annotation file. Pseudogenes (type 5) are generally derived from a non-synonymous mutation that partially or totally truncates a protein. In these cases, the presence of an in-frame start codon downstream of the mutation can give rise to a fragment of the original gene sharing its properties. To detect such cases, RanSEPs searches for cases where a SEP in the reference genome (gene A') was near a downstream or upstream gene (gene A) and these two together (gene A-A') were homologous to a single gene in any of the closely related species. In this case, gene A' would be labeled as a potential pseudogene.

## RanSEPs methods

RanSEPs implementation is fully based on Python (version >2.7.x), using functions included in and tested in the scikit-learn package (Pedregosa *et al*, 2013). A fully functional version of RanSEPs is documented in and downloadable from GitHub and http://ranseps. crg.es/.

*Set definition*
In this step, it is important to define closely related organisms in the database to avoid an overestimation of conserved smORFs. This process is automatically performed by RanSEPs after evaluating the complete conservation database. The non-conserved smORFs are randomly and iteratively sampled with the selected set size. For the positive set, a minimum size of 100 true proteins is required. Although it is preferred that this set includes all the annotated SEPs of the organism, the user can define the specific percentage of SEPs that are included.

## Protein feature computation

Complex featurization of sequences was performed using the Python package *propy* (Cao *et al*, 2013). This package computes more than 1,500 features for each single sequence, covering protein attributes like amino acid composition, dipeptide and tripeptide composition, Moreau–Broto, Moran, and Geary autocorrelations, sequence-order-coupling number, and physicochemical properties. Importantly, as many of these features present a high correlation, including all of them could strongly over fit our training and test sets. To avoid this problem, we ran a Recursive Feature Elimination (RFE) to prune the least important features from the trees (i.e., features that do not efficiently separate positive and negative sequences). We applied this approach over the 109 organisms and selected the three best features by average: quasi-sequence-order-coupling numbers based on the Schneider–Wrede physicochemical distance matrix, hydrophobicity, and secondary structure.

RF classification enables features to be sorted by their importance and then compares these weights in a quantitative manner. Taking this into account, we added several sequence attributes of specific biological interest to the comparison of coding features between microorganisms. These are as follows:

*Start codon:* The ATG start codon is prevalent over alternative start codons like GTG and TTG. To consider this effect in the classification, we assigned a binary classification where 0 represents annotations that do not have an ATG codon in their first 5 codons, and 1 represents otherwise.

*GC content:* GC content is computed as the count of G + C divided by the length of the annotation. As described, GC content has a direct effect on the probability of finding start and stop codons.

*Ribosome-binding site (RBS) stacking energy:* RBSs are important elements in translation regulation in some bacterial species. As motifs associated with this element can vary between organisms, we represented the stacking energy of the −15 to the start codon window. This value is close to −1.26 of free energy in the presence of the AGGAGG motif.

*Ribosome presence:* Ribosome presence is included as a binary value where 1 indicates the presence of any of the possible Shine–Dalgarno sequences known to act as an RBS (Mir *et al*, 2012).

*−10 + 20 stacking energy:* Multiple studies suggest that specific sequence requirements at the 5′ end of an mRNA impact translation efficiency. In the same way as for RBS, we computed the stacking energies for the 30 bases spanning the −10 to +20 region (with respect to the start codon).

Special features are measured in a relative manner using a "feature" set that is sampled from the positive set (same properties) but not used in the training process. Features extracted from this set are as follows:

*−10 score and +20 score:* scores computed for the separate elements based on a position weight matrix (PWM) of those regions computed from annotated genes of the feature set.

*Hexameric measures:* calculated by sliding a 6-base window along the sequence, starting in frame with the annotation (dicodon frequency), +1, +2, and the combination of all the possible hexamers (n hexamer measure). For each sequence, a single value per frame and in combination is extracted. This value is computed as the logarithmic odds ratio between the observed hexamer frequencies and the expected one computed from the feature set. Both sets of frequencies were normalized by the background frequencies based on the GC content.

*Codon adaptation index (CAI):* a measure of the deviation of codon presence in a sequence from a background model that is extracted from the feature set of proteins. By implementing this measure in addition to the hexameric measures, we take into account synonymous codons (Sharp & Li, 1987).

*2 amino acids, N and C terminal:* two features representing the importance of specific amino acids at the initiation and termination sites. The importance of these features depends on the species.

## RF tuning calibration

After defining the types of sequences to include in each set, we exhaustively explored the parameter space to properly calibrate the single classifiers. RanSEPs presents two levels of complexity in its tuning, single classifiers and a global classifier, where the latter is the combination of single RFs. Tuning of single classifiers was performed in an exhaustive manner, iterating and testing every combination between: (i) 2, 4, 8, 16, 32, 64, 128, 256, 512, and 1,024 trees; (ii) 10- and 25-fold cross-validation; (iii) test sizes of 1, 5, 10, and 20%; (iv) positive, negative, and feature set sizes between 100, 200, and 300 sequences; (v) percentage of SEPs in each set between 0, 5, 10, 25, and 50%; (vi) maximum depth of the forest between 0, 10, and 20; and (vii) minimum samples per leaf from 1 to 20. For each combination in that parameter space, we combined 5, 10, 20, 30, and 50 single classifiers into the global classifier. We then tested their accuracies based on the AUC of their ROC curves to find the best parameters using the same test size combinations of single RF in a global manner.

In the end, we ended up with the default configuration shown in Table 2 for *M. pneumoniae*. This set of parameters worked properly in organisms with < 100 annotated SEPs and genome sizes < 1 kilobase. In the case of organisms with multiple SEPs (> 100) already annotated in NCBI and a bigger sized genome (> 1 kilobase), we observed more adjusted predictions (an equal TPR but a lower FPR) when increasing the negative set size to 2,000, and 85% of SEPs in the positive/feature set with size equal to 200. These rules are implemented in RanSEPs as automatic considerations. RanSEPs can run as a general random forest algorithm (1 classifier) with regular k-fold cross-validation procedures or generating multiple classifiers with randomized training sets to provide an averaged probability.

**Table 2.  RanSEPs default settings.**

| | |
|---|---|
| Positive set size | 100 |
| Negative set size | 500 |
| Feature set size | 100 |
| Percentage of SEPs in positive and feature set | 25 |
| Number of single classifiers per general classification | 5 |
| Number of trees | 100 |
| Maximum depth | 0 |
| Minimum samples per leaf | 5 |

Parameter configuration used for the detection of proteins in *Mycoplasma pneumoniae*.

### Feature weight estimation

An out-of-bag (OOB) approach was implemented to compute the importance of each feature in the classification task. This algorithm works by leaving a group of labeled points that will be classified out of the training set. For each classification, the algorithm permutes a feature while leaving the rest unchanged, and measures the error increase comparing the labels with the classes assigned.

### RanSEPs output

RanSEPs output includes several files related to coding-potential features and classification stats, in addition to the classification task results (Dataset EV23). An additional "parameters.txt" file is generated in order to keep track of the parameters used in each specific execution.

### Validation set definition

The positive set (*n* = 570) comprises 307 SEPs detected by MS with ≥ 2 UTPs from the six Mycoplasma species considered, *Escherichia coli, Pseudomonas aeruginosa,* and *Staphylococcus aureus* (Datasets EV2, EV3, EV6–EV13). Second, we performed searches in six additional public datasets for *Lactococcus lactis*, *Helicobacter pylori*, and *Synechocystis*, extracting a total of 166 SEPs (Datasets EV14–EV16; Lahtvee *et al*, 2011; Müller *et al*, 2013; Gao *et al*, 2015). These two sets together (*n* = 473) not only included multiple annotated proteins from the organism used as a reference (*n* = 335) or in a closely related one (*n* = 87), but also 25 potentially new SEPs that were not previously annotated in the corresponding reference genomes or other organisms of the RanSEPs database (Dataset EV17). Third, six SEPs detected by targeted MS (MS1 Targeted Area Extraction) using C13C(6)15N(2)-labeled peptides from the *M. pneumoniae* proteome (Dataset EV4). Fourth, 97 previously reported SEPs from six different bacteria, well-characterized in the literature and experimentally detected (Hemm *et al*, 2010; Kodama *et al*, 2011; Storz *et al*, 2014; Baumgartner *et al*, 2016; Duval & Cossart, 2017; Impens *et al*, 2017; VanOrsdel *et al*, 2018; Dataset EV18).

The negative set (*n* = 570) was extracted from two different sources. First, we randomly selected 556 SEPs from a collection of putative SEPs satisfying the following criteria: (i) ≥ 2 HR_UTPs by PeptideSieve; (ii) no NUTP/UTP signal by MS; and (iii) not conserved in any closely related bacteria (highest e-value > 0.01 by BLASTP). This set was balanced to be comparable with the positive set (the same average amino acid length (35 aa)) and to be representative of the 12 bacterial species considered (Dataset EV18). Additionally, we included 14 SEPs detected with 1 UTP but not detected by C13 proteomics: 2 found in *M. pneumoniae* and 12 in *Helicobacter pylori* (Friedman *et al*, 2017).

### Annotation tool comparative

As quality metrics for the prediction, we used the accuracy (rate between true positives and true negatives over the total number of tested SEPs) and the AUC between true-positive and true-negative rates (the closer to 1 the better). AUC was measured by ROC curves, and accuracy was supported by precision–recall curves. All searches for validated SEPs using RanSEPs were performed excluding the target proteins of the training process. To run BASys predictions, we used their web service (basys.ca) and selected the arguments: gram-positive/negative and providing specific CDS nucleotide sequences of each target organism to perform a customized search. A CPC search was performed at their website (cpc2.cbi.pku.edu.cn) using the default general search, providing each target genome putative ORFs. GeneMarkS (exon.gatech.edu/Genemark/genemarks.cgi) was run using the default search and selecting the TGA option as a Tryptophan codon for Mycoplasma species. To predict genes with Glimmer, we used the desktop version 3.0 downloaded at ccb.jhu.edu/software/glimmer/. In order to adjust the search for predicting small proteins in each organism, we specifically defined the use of start codons with custom probabilities based on their recurrence in annotated genes (e.g., *M. pneumoniae*: ATG = 0.86, GTG = 0.073, and TTG = 0.067). Additionally, we set a minimum size of 10, and trained the search with the annotated genes of each specific organism excluding the target proteins. To make the comparative meaningful, we standardized the metric provided by Glimmer to a probability scale of 0–1. The last software, Prodigal (github.com/hyattpd/Prodigal), was used as a desktop application forcing a full motif scan of Shine–Dalgarno subsequences, and using the annotated genes of each specific organism excluding the target proteins as a reference.

### Functionality studies

Based on their described function in NCBI, the annotated SEPs from the 109 bacterial species were assigned to nine functional categories (Dataset EV18). Functions assigned by homology inference were not taken into consideration. Annotated SEPs with known functions were used as the query database to assign functions by homology to the remaining putative SEPs with undefined functions. Homologous gene pairs were defined using the same e-value, aligned length, and shared size thresholds as in the other analyses.

The desktop version of Phobius (http://phobius.sbc.su.se/) was used to predict any signal peptides and transmembrane segments in our predicted SEPs using default settings and only differentiating between gram positives and negatives.

## Data and software availability

- An index of the datasets presented has been included in the Appendix.
- RNA-Seq datasets: ArrayExpress E-MTAB-6203 (https://www.ebi.ac.uk/arrayexpress/experiments/E-MTAB-6203)
- Proteomics datasets: PRIDE PXD008243, PRIDE PXD010490, PXD011038 (https://www.ebi.ac.uk/pride/archive/projects/PXD000208; https://www.ebi.ac.uk/pride/archive/projects/PXD010490; https://www.ebi.ac.uk/pride/archive/projects/PXD011038)
- RanSEPs (http://ranseps.crg.es/)

**Expanded View** for this article is available online.

### Acknowledgements

We thank Dr. Luca Cozzuto from the Bioinformatics Unit at CRG for providing valuable guidance for the conservation studies. Also, we would

like to thank Dr. Carolina Gallo, Dr. Eva Yus, and Dr. Raul Burgos for providing MS data. Finally, we thank Dr. Marc Weber for his recommendation about how to analyze Ribo-Seq data. We acknowledge support of the Spanish Ministry of Economy, Industry and Competitiveness (MEIC) to the EMBL partnership, the Spanish Ministry of Economy and Competitiveness, "Centro de Excelencia Severo Ochoa", the European Research Council (ERC) under the European Union's Horizon 2020 research and innovation program under agreement No 670216 (MYCOCHASSIS), the CERCA Programme/Generalitat de Catalunya, the European Regional Development Fund (ERDF) project from Instituto Carlos III (ISCIII, Acción Estratégica en Salud 2016; reference CP16/00094), and "Secretaria d'Universitats i Recerca del Departament d'Economia i Coneixement de la Generalitat de Catalunya" (2014SGR678). The CRG/UPF Proteomics Unit is part of the "Plataforma de Recursos Biomoleculares y Bioinformáticos (ProteoRed)" supported by grant PT13/0001 of Instituto de Salud Carlos III from the Spanish Government.

## Author contributions

SM-V performed computational and statistical analyses, and methodology assessment, developed RanSEPs, and interpreted results. TF performed sample preparation for RNA-Seq and MS experiments, wrote the methodology, and corrected and improved the final version of the manuscript. GE-G processed the proteomics samples, wrote the methodology followed, and, together ES, provided valuable discussion about MS results interpretation. RM performed sample preparation for MS experiments and wrote methodology. AG created the webpage where RanSEPs program and results are located. LS provided direct supervision, interpreted results, and helped design this research. ML-S designed the experimental approach, interpreted results, and provided direct supervision of the research. SM-V and ML-S performed the functional study and created the figures and tables. SM-V, LS, and ML-S wrote the manuscript. All authors read and approved the final manuscript.

## Conflict of interest

The authors declare that they have no conflict of interest.

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
