## [Review Process File · Molecular Systems Biology]

Unravelling the hidden universe of small proteins in bacterial genomes

Samuel Miravet-Verde, Tony Ferrar, Guadalupe Espadas-García, Rocco Mazzolini, Anas Gharrab, Eduard Sabido, Luis Serrano and Maria Lluch-Senar.

Review timeline:

Submission date:	27 th February 2018
Editorial Decision:	2 nd May 2018
Revision received:	18 th September 2018
Editorial Decision:	18 th December 2018
Revision received:	11 th January 2019
Accepted:	21 st January 2019

Editor: Thomas Lemberger

Transaction Report:

1st Editorial Decision

2nd May 2018

Thank you again for submitting your work to Molecular Systems Biology. We have now heard back from the three referees who agreed to evaluate your manuscript. As you will see from the reports below, the referees find the topic of your study of potential interest. They raise, however, substantial concerns on your work, which should be convincingly addressed with further experimentation and analyses.

The major concerns raised by the reviewers refer to the need of more solid, orthogonal evidence for the existence of a "significant subset of the small proteins reported" and an assessment of false/true positive rates.

If you feel you can satisfactorily deal with these points and those listed by the referees, you may wish to submit a revised version of your manuscript. Please attach a covering letter giving details of the way in which you have handled each of the points raised by the referees. A revised manuscript will be once again subject to review and you probably understand that we can give you no guarantee at this stage that the eventual outcome will be favorable.

REFeree REPORTS

Reviewer #1:

This study represents a significant amount of work, and the identification of unannotated small open reading frame-encoded proteins is important. However, given the potential artifacts associated with the identification and characterization of small proteins due to their low information content, the authors need to be much more careful about their analyses, the presentation of the experiments and data, and their conclusions. A flood of insufficiently substantiated small protein data could do more harm than good.

1. A key component of the authors' analysis is the supposed validation of proteins by mass spectrometric analysis. Given the difficulties in mass spectrometric analysis of small proteins in particular (proteins with two or more UTPs could also be artifacts), the authors need to provide independent evidence of small protein synthesis, such as immunoblot detection of tagged proteins expressed from the chromosome, for a significant subset of the proteins reported.
2. The authors are not sufficiently explicit about experimental details and results. As just two examples, the conditions and mutants for the 116 shotgun mass spectrometric experiments should be easy to decipher, and the data for all 109 bacterial smORFomes should be easily accessible (www.ranseps.crg.es was not available).
3. In the absence of any experimental tests, the authors need to be more critical of the output of their computational analysis of the small protein features and functions. A problem inherent in small proteins is the low information content which makes hydrophobicity and homology searches tenuous. How well do their analyses perform on known and random data sets? For example, I question whether the authors can really distinguish between signal peptide sequences and transmembrane regions to provide such specific numbers (9.7% and 15%)?

More minor comments:

4. Page 3, line 8: Since the small proteins being described appear to be encoded by specific genes rather than being cleaved from larger proteins, they should be referred to as proteins rather than peptides.
5. Page 6, lines 2-4 from bottom: These two sentences are contradictory "In fact we found 19 decoy SEPs with {greater than or equal to} 1 UTP...All decoy proteins were not detected when considering {greater than or equal to} 1 UTP.
6. Typographical error: "aand"

Reviewer #2:

The characterization of short open reading frames (smORFs) that code for small proteins has become a very interesting and important topic. Unfortunately, most gene assembly and annotation tools ignore this arena, and thus this important category of molecules is often unknown. To address this issue, this manuscript reports an approach to combine a novel bioinformatics tool (RanSEPs) with omics measurements to examine numerous bacterial smORFomes. They find that about 25% of proteins in a bacterium are SEPs, and assign functional categories to many of them.

In general, this paper is scientifically sound and presents interesting results. The layout and discussion are logical and systematic. Numerous comments are given below:

1. The text construction is very poor. There are numerous grammar problems, several typos, and the text in many places is quite difficult to wade through. Serious revision and clarification is needed in many places.
2. Page 5, line 4 - there are other informatic methods that have been/are used for SEP identification. The authors need to also put their tool in perspective with respect to the other methods. They do this later in the manuscript, but the reader is left hanging here about what is unique and ground-breaking about their method?
3. Page 5, line 9 - what does "a costless manner" mean? The authors need to provide some additional metrics (speed, accuracy, etc.) for their tool.
4. Page 5, line 13 - the observation of a much larger proportion of SEPs in bacterial genomes immediately raises the concern of "noise" in the detection. The authors are attempted to use experimental validation (which is good), but that does not provide evidence in each case. Maybe the

authors can be a bit more specific here about control of false positive/false negative parameters and expectations in RanSEP.

5. Page 6, line 8 - the "decoy dataset" is confusing and needs to be clarified a bit in this text section. It is detailed a bit better in the Materials section, but not clear here.

6. Page 6, bottom - as the authors point out, the > 2 UTP requirement may be too stringent for these small proteins. Why not consider 1 UTP plus one "other" peptides, whether it is unique or non-unique? This is more consistent with how much of the proteome field uses thresholds. The requirement of > 2 UTP is too restrictive. Was mass accuracy enforced on these peptides? That would provide additional restrictions.

7. Page 7, bottom - this discussion of RanSEP is nice and somewhat offsets my comment #4 above, but there is still some confusion about how TPR and FPR are set and validated?

8. Page 9, bottom - the authors should be careful here in the header to state that "RanSEP is the only tool..." There are other available, and have been demonstrated on other systems, including plants.

1st Revision - authors' response

18th September 2018

Experimental approaches:

Mass spectrometry (MS) data

- Criteria to identify proteins: in addition to our first threshold of ≥ 2 Unique Tryptic Peptides (UTPs) for accepting a SEP as positive, we have now explored, as suggested by one of the reviewers, a different criteria: 1UTP - 1 not unique (NUTP).
- New data for other bacterial species: we have extended the bacterial organisms studied by mass spectrometry (MS) from the original 6 Mycoplasmas species to an additional 6 species.
 - We have looked for SEPs in total cell extracts of *Escherichia coli*, *Pseudomonas aeruginosa* and *Staphylococcus aureus*. Briefly, after running tricine SDS gels to recover SEPs, we cut a band corresponding to < 10 kDa and extract the proteins for MS.
 - Also, we have re-analyzed (with the same parameters as in the self-generated datasets) publicly available MS datasets generated to detect SEPs and reported in the literature for: *Lactococcus lactis* (PRD000266), *Synechocystis* sp. PCC6803 (PXD001246) and *Helicobacter pylori* (PXD000054).

Independent of MS:

- We have used ribosome profiling in *E.coli* and *M. pneumoniae* to support the claim that two or more UTPs unequivocally confirm the existence of a SEP, a claim that was questioned by one of the referees.
 - In the case of *E. coli*, we processed and analyzed a publicly available dataset reported by Hücker *et al.* 2017 (SRA accession: SRP113660) to detect novel SEPs. Study of correlations between proteins detected by MS and proteins identified by ribosome profiling corroborate that the filter of 2 UTPs is the most accurate filter for identifying a protein by MS.

- Also, datasets for *M. pneumoniae* have been generated in our group. We plan to publish these in an independent manuscript. However, these datasets have been used to provide additional support to the results observed in *E. coli*.

Computational approaches:

- During the review process we have improved specific points pertaining to the functioning and availability of RanSEPs :
 - The first version required the genome sequence in fasta format and the CDS in a multi-fasta file. Now RanSEPs accepts genbanks as input file so that only one single file is required with the genome and coding sequences in addition to gene features information.
 - Previously, RanSEPs was using a predefined database. The current version accepts user-defined databases. This change makes RanSEPs more flexible and users will have more control over how the predictions are computed.
 - The first version required the database of putative ORFs to be generated and a conservation study to define the negative training set. This step was redundant when multiple predictions were required for the same bacterial species (e.g., testing different parameters). RanSEPs now accepts these required files as arguments, eliminating unnecessary computational steps and saving computational costs.
 - RanSEPs predictions are now supported by conservation studies to detect:
 - Novel SEPs in an organism of interest with function/annotation in a different species.
 - Pseudogenes and repeated sequences that would count as false positives.
 - Re-design of www.ranseps.crg.es to include an 'about' section with the main description of the approach and the possibility to download the program directly (as opposed to downloading it from the GitHub repository).
- The main efforts have focused on improving the validation and assessing the quality of the predictions. Originally, we compared the predictions between RanSEPs and 5 other tools for 38 SEPs detected in 6 Mycoplasmas and 97 SEPs described in the literature (135 in total) as a positive validation set and a balanced negative set from *M. pneumoniae*. We now include::
 - New validation sets: The positive set has been extended from 135 to 570, after including all the SEPs detected with ≥ 2 UTPs for 6 additional bacterial species that we studied by MS. The negative set has been defined using conservation

analysis and peptide responsiveness as a reference. The negative set is not limited to *M. pneumoniae* anymore and all the species studied are considered including a balanced and randomly selected set of SEPs satisfying: no MS signal, no conservation in closely related species and having at least 2 potentially high responsive peptides.

- To evaluate the rate of false positives we used the previous set of negative SEPs without random sampling (~15,000 entries).
- We re-calculated the prediction quality metrics for RanSEPs and the rest of the software tools using the previous sets as a reference.
- A computational time cost comparative study has been included in the article.
- CPC predictions used in the comparative analysis have been updated to include the scores provided by the new version CPC2. The first version of the program (desktop and web server) is currently out of service and we could not run it for the updated validation test. Consequently, we decided to recompute the results to ensure reproducibility.
- An accident occurred with the ESPPredictor server that was used to predict responsive peptides and no functional desktop version was found (more information). As we required additional predictions of UTP responsiveness, we computed all the results with a different software: PeptideSieve. Manuscript and related figures were updated accordingly. These new approaches did not change the conclusions extracted in this study.

We also showed that external tools used to predict SEPs features (PeptideSieve for peptide responsiveness, Phobius for signal peptide prediction and BlastP for function prediction) do not present biases derived from the size of the evaluated annotations.

Referee 1

This study represents a significant amount of work, and the identification of unannotated small open reading frame-encoded proteins is important. However, given the potential artifacts associated with the identification and characterization of small proteins due to their low information content, the authors need to be much more careful about their analyses, the presentation of the experiments and data, and their conclusions. A flood of insufficiently substantiated small protein data could do more harm than good.

We thank the reviewer for pointing out the importance of identifying new SEPs and we agree with him/her that their characterization, as well as, identification is technically and computationally challenging. For this reason, we performed the integrative study of transcriptomics and proteomics data showing the experimental limitations and the importance of developing new computational approaches such as RanSEPs that can help in prioritizing which SEP candidates to study experimentally.

1. A key component of the authors' analyses is the supposed validation of proteins by mass spectrometric analysis. Given the difficulties in mass spectrometric analysis of small proteins in particular (proteins with two or more UTPs could also be artifacts), the authors need to provide independent evidence of small protein synthesis, such as immunoblot detection of tagged proteins expressed from the chromosome, for a significant subset of the proteins reported.

We understand the concern of the reviewer about mass spectrometry data. We would like to clarify that the positive set (n=135) included 97 experimentally validated SEPs collected from the literature, most of which were identified by targeted and integrative approaches, not only by label free proteomics. In addition, we showed that the threshold of two or more unique peptides (≥ 2 UTPs) is strict enough to avoid false positives (validated by targeted proteomics, first section of results, 3rd paragraph). We apologize that it was not properly described in the manuscript, and have now tried to explain it better in the current version. However, we agree with the point made by the reviewer: "more evidences of small protein synthesis were required and they should be independent of MS". To address this issue we carried out two different approaches, a MS-dependent and a MS-independent approach.

In the first approach, we aimed to validate SEPs present in the proteome of other bacteria species by MS. First, we performed new MS experiments enriching for SEPs: 2 samples from *Escherichia coli*, 4 from *Pseudomonas aeruginosa* and 4 from *Staphylococcus aureus*. In addition, we included 6 MS searches from public databases coming from experiments specifically designed to detect SEPs. This selection included samples coming from *Lactococcus lactis* (3 samples), *Helicobacter pylori* (32 samples) and *Synechocystis* (24 samples). Altogether, the SEPs collected from our new experiments, the bibliography, and targeted proteomics allowed us to define a positive set of 570 experimentally detected SEPs. We then used this set for validation and for comparing different tools, with RanSEPs providing the best predictions (AUC=0.95, Accuracy=0.89). This analysis replaces the previous validation section in the manuscript; we have added that together multiple assessment metrics: sensitivity, specificity, AUC, and accuracy and supported the results with a ROC curve and Precision-Recall visualizations (Figure 3C and Appendix S7, Datasets EV18-EV19).

In the second approach, we used MS-independent experimental evidence to validate the cutoff of 2 UTPs as a way to unequivocally decide that a SEP is real. After evaluating the suggestion made by the reviewer to perform immunoblot detection of tagged proteins expressed from the chromosome, we thought that this technique was not the most appropriate and instead decided to apply Ribosome profiling combined with ultra-sequencing (RiboSeq). The reasons against using immunoblot detection were:

1. Detection by immunoblot requires the addition of a tag to the proteins. This tag could affect protein stability and half live since the main degradation pathway of proteins in *M. pneumoniae* is through Lon protease. This protease recognizes the C-terminus of proteins, among other signals. Furthermore, including the tag in the genome requires targeted modifications by double crossover that is dependent on homologous recombination. For the time being, homologous recombination cannot be used as tool to engineer the genome of

M. pneumoniae. Thus, including the tag at the 5' or 3' end of the endogenous copy of the smORF in the chromosome is not feasible. Alternatively, we could include an extra copy by transposon mutagenesis. However, the problem is not only that the SEP would be randomly inserted (depleting some genes), but that many of the predicted SEPS are inside operons. Therefore, you would need to merge the promoter of the first operon of *M. pneumoniae* to the SEP. In this bacterium, we know that the combination of the 5' UTR with the first 20-30 nucleotides could result in combinations that prevent translation. Non published results obtained in our lab (Figure R1) showed that to express heterologous proteins in *M. pneumoniae*, a Mycoplasma promoter and the first 20 bases of a Mycoplasma gene need to be added.

Figure R1 (a) Schematic representation of the different fusion products between the S200pmp gene and the Tet repressor coding sequence. The construct without fusion had directly the upstream promoter region of S200pmp before the ATG codon of the Tet repressor coding sequence, which would include any 5' UTR sequence. (b) Western blot of strains transformed with the different constructs shown in (a) On the right side of the plot a sample of *M. pneumoniae* M129 wild type (wt) is shown as control. The expected size of the Tet repressor is 37 kDa and 40 kDa for the longest fusion. While the specific bands run slightly above the expected size they show the expected pattern of increasing size with increasing fusion length.

- It has been recently reported that the success rate of detection of SEPs by immunoblot in *E. coli* is very low (45%) (Van Orsdel *et al.* 2018). In this study by Van Orsdel *et al.*, an epitope tag was added to the 3' end of 80 smORFs on the chromosome. However, SEP synthesis could only be confirmed by immunoblot assays for 36 (a 45% success rate). These selected smORFs had preliminary evidence for their existence and represented diverse sequence characteristics, including conservation, predicted transmembrane domains, smORF direction with respect to flanking genes, ribosome binding site (RBS) prediction, and ribosome profiling results. This low success rate reflects the experimental limitations of this technique. These 36 SEPs are used in our validation positive set and it is interesting to remark that 28 of them (77%) presented ≥ 2 UTPs. This therefore reinforces the reliability of the criteria used in our analyses.

3. Additionally, endogenous promoters could be transcriptionally regulated and the expression of SEPs could depend on specific conditions. For example, the alternative sigma factor MPN626 and its targets are not seen under normal growth conditions in *M. pneumoniae*.

For all these reasons, we decided to study the correlation between ribosome profiling and detection of proteins by MS. As we mentioned in the introduction of the manuscript, we are aware that RiboSeq is not the best technique to detect SEPs, since the binding of the ribosome does not indicate the frame in which the mRNA is translated. However, this technique has been shown to be reliable in detecting SEPs located in intergenic regions (no overlap with known annotations) in a high-throughput manner in *E. coli* (Hücker *et al.* 2017) and *Synechocystis* (Baumgartner *et al.* 2016). Following these observations, we processed the raw dataset presented in Hücker *et al.* 2017 for *E. coli* (3 experimental conditions) and selected non-overlapping annotations presenting ≥ 2 high responsive UTPs by PeptideSieve. Ribosome profiling results showed that the mRNA of SEPs, detected with ≥ 2 UTPs by MS presents significantly more ribosomes bound than those detected with 1 UTP and not detected by MS in *E. coli* (Figure R2A, below; Appendix Figure S5A in manuscript). Interestingly, the difference in the RCV values of SEPs split by number of UTPs is equal to the observed values for 1,039 standard size proteins already validated and annotated in the reference genome (Figure R2B, p-value=0.03). Additionally, in our lab, we performed this same approach using a dataset from *M. pneumoniae*. The results obtained in *M. pneumoniae* are in agreement with those observed in *E. coli* (Figure R3: not shown in the published Review Process File). These experiments will be presented in an independent manuscript and as such, they are not included in this manuscript.

Figure R2. RCV comparative for SEPs (A) and standard size proteins (B) in *E. coli*.

Figure R3: Figures for Referees not shown in the published Review Process File.

These results indicate that SEPs selected with our criteria (≥ 2 UTPs) are similar in terms of RCV to annotated and characterized standard size proteins, supporting their existence and their use as positive set to validate in RanSEPs based on MS data. This analysis is in the first results section (6th paragraph in ‘Key factors and criteria for the experimental identification of SEPs’) and described in the Appendix Supplementary Methods material and figures (‘Ribosome profiling in *Escherichia coli*’ section and Appendix Figure S5).

2. The authors are not sufficiently explicit about experimental details and results. As just two examples, the conditions and mutants for the 116 shotgun mass spectrometric experiments should be easy to decipher, and the data for all 109 bacterial smORFomes should be easily accessible (www.ranseps.crg.es was not available).

We apologize for possible difficulties during the revision process due to a lack of or inaccessible information. We have improved the descriptions and information by:

- Adding a description of each mutant or condition for the different shotgun experiments in an additional column in the Supplementary file (Dataset EV2).

- Making accessible the data for all the 109 bacterial smORFomes in the webpage www.ranseps.crg.es (login requirement will be removed upon publication; username: reviewer and password: reviewerCRG123456). This was previously unavailable due to privacy requirements.

- Thoroughly reviewing the content in the Supplementary tables so that it is more self-explanatory and does not require merging of information from different tables:

- NCBI description added to 6 ORF databases when protein found annotated in Datasets EV1, EV6-EV16.

- New Dataset EV17 including all the SEPs detected in this work, including all the information required to validate the observation (UTPs detected, sequences, annotation, RanSEPs scores, etc.).

3. In the absence of any experimental tests, the authors need to be more critical of the output of their computational analysis of the small protein features and functions. A problem inherent in small proteins is the low information content which makes hydrophobicity and homology searches tenuous. How well do their analyses perform on known and random data sets? For example, I question whether the authors can really distinguish between signal peptide sequences and transmembrane regions to provide such specific numbers (9.7% and 15%)?

We agree with the reviewer that we lacked proper validation for the results exposed in the functional characterization of predicted SEPs. In the reviewed version, we have improved the validation of the results obtained by the three external tools used: PeptideSieve, BlastP and Phobius.

- PeptideSieve was used to predict the number of responsive UTPs in order to estimate whether or not to expect the detection of a protein by MS (Mallick et al, 2007). This tool is

independent of the annotation length as it *in silico* digests proteins and predictions are computed over the resulting UTPs if they are longer than 5 amino acids in length. Therefore, probabilities computed by PeptideSieve only depend amino acid composition of the UTP whilst the protein has no impact in the prediction.

- BlastP is used in several steps of our work: definition of negative sets for RanSEPs, prediction of pseudogenes and prediction of functions. We implemented multiple conditions for considering homology between SEPs in order to ensure that these predictions were not biased by the size of the annotation being explored (explained in Material and Methods, ‘Conservation analyses’ section). As an additional test to address the reviewer’s concerns, we repeated the prediction of functions with the ‘decoy’ dataset used in MS, including randomly generated sequences, and found that none of them passed the threshold used in the case of predicted SEPs. This has been included in the manuscript at the end of the 2nd paragraph in the last section of the Results “Functional assessment to novel SEPs”:

‘We repeated this search with the dataset of ‘decoy’ proteins used for MS as the target, and found that no sequence passed the thresholds required to be considered homologous. As such, we would not expect to have false positives by chance.’

- Finally, we did not use RanSEPs to predict proteins with signal peptides or transmembrane proteins, but used Phobius (Käll *et al.*, 2004). To test the efficiency of Phobius in predicting transmembrane segments and signal peptide presence, we performed the same test as in the previous paragraph and found that less than 1.2% of the sequences were predicted to have to these (sample size=20,100 SEPs). This shows that the percentage observed for predicted SEPs (9.7%) was higher than what we would expect by chance. In addition, we performed another test where we subsetting annotated proteins of *M. pneumoniae* with a size >200 aa and a signal peptide. Then, we sequentially shortened their C’- terminus. This was done to check for a loss in the accuracy of predicting peptide/transmembrane segments due to protein size. This analysis showed that Phobius works as expected for SEPs and thus we are able to trust its predictions. This information is presented in the manuscript at the end of the last paragraph in the last section entitled “Functional assessment to novel SEPs” in Results:

‘The percentage of SEPs with a signal peptide was higher than expected by chance when compared with the same ‘decoy’ set of SEPs used in MS (9.7% for predicted SEPs, 1.2% for ‘decoy’ SEPs, unpaired two-tailed t-test p-value=0.018). Moreover, to confirm that the results obtained with Phobius are meaningful with regards to SEPs, and that protein size did not bias the analysis, we ran a test over a set of annotated standard proteins in which we sequentially shortened their C-terminus. The sensitivity of Phobius is higher than 80% for sequences over 30 amino acids. For sequences under 30 amino acids, however, we see values lower than 50%; this is expected considering that Phobius specifically searches for a motif presented by the 16 to 30 amino acids of the N-terminus of a protein. If the motif is located within the first amino acids and is short, Phobius still detects the proteins as positives (see Appendix Supplementary Methods, Appendix Figure S11).’

This analysis has been extensively detailed in a last section of the Appendix Supplementary Methods information file and Appendix Figure S14 has been added to represent the results.

4. Page 3, line 8: Since the small proteins being described appear to be encoded by specific genes rather than being cleaved from larger proteins, they should be referred to as proteins rather than peptides.

We agree with the reviewer and we have changed the terminology accordingly. In the SEPs description we have now explained the term as small encoded proteins instead of peptides.

5. Page 6, lines 2-4 from bottom: These two sentences are contradictory "In fact we found 19 decoy SEPs with {greater than or equal to}1 UTP...All decoy proteins were not detected when considering {greater than or equal to}1 UTP.

The whole first section of results about key factors and determinants in the detection of SEPs has been edited to include a suggestion from the second referee and we thoroughly checked the numbers to avoid similar mistakes to the one pointed out by the referee.

6. Typographical error: "aand"

We apologize for the error that is corrected in the current version of the manuscript. This version has been externally reviewed by a native English scientific editor to improve the quality and clarity of the text.

Referee 2

The characterization of short open reading frames (smORFs) that code for small proteins has become a very interesting and important topic. Unfortunately, most gene assembly and annotation tools ignore this arena, and thus this important category of molecules is often unknown. To address this issue, this manuscript reports an approach to combine a novel bioinformatics tool (RanSEPs) with omics measurements to examine numerous bacterial smORFs. They find that about 25% of proteins in a bacterium are SEPs, and assign functional categories to many of them. In general, this paper is scientifically sound and presents interesting results. The layout and discussion are logical and systematic.

We are very grateful to the reviewer for his/her positive comments and for recognizing the impact of our work. We expect to open new lines of research in Biology orientated towards the characterizing and better understanding the function of these newly discovered SEPs.

Numerous comments are given below:

1. The text construction is very poor. There are numerous grammar problems, several typos, and the text in many places is quite difficult to wade through. Serious revision and clarification is needed in many places.

We apologize for the misspellings and grammar errors. The current version of the manuscript has been externally reviewed by a native English scientific editor to improve the quality and clarity of the text.

2. Page 5, line 4 - there are other informatic methods that have been/are used for SEP identification. The authors need to also put their tool in perspective with respect to the other methods. They do this later in the manuscript, but the reader is left hanging here about what is unique and ground-breaking about their method?

We have added the following sentence to the introduction to highlight the RanSEPs' potential and specificities which make it better for SEPs prediction (last sentence 4th paragraph):

“The better prediction accuracy of our method is due to the iterative randomization of the training set, a technique that enables the capturing of additional protein-related information during training. In addition, as the training set is biased to include more SEPs, it places a higher level of importance on the possible alternative features in the classification.”

3. Page 5, line 9 - what does "a costless manner" mean? The authors need to provide some additional metrics (speed, accuracy, etc.) for their tool.

That sentence was considering RanSEPs a “costless manner” in comparison to experimental approaches. Nonetheless, we understand it was not well explained in the text and have now removed it.

A comparison of computational performance has been included in the Appendix Supplementary Methods document. In this comparison we run the same tests for RanSEPs, glimmer and prodigal. CPC, BASys and GeneMarkS were not considered as they are web services where queue systems are required to run their applications and this provides an inaccurate representation of their performances.

The test performed consisted in running each tool with 8 bacterial genomes ranging in genome size from 0.5Mb to 9.1Mb. With this approach we showed that the running time of RanSEPs per prediction is comparable to the ones presented by glimmer and prodigal. However, time performance of our tool increases with the number of iterations selected by the user; this additional cost can be justified by the significant increase in SEPs prediction accuracy. In terms of CPU, the three tools presented a similar percentage of CPU use.

In the manuscript we exposed this analysis in the Appendix Supplementary Methods information, as well as, in the Appendix Figure S8.

In terms of accuracy, we have significantly improved the validation and method comparative section by increasing the number of entries in the validation set. This analysis replaces the previous validation section in the manuscript; we have added that together multiple assessment metrics for the 6 tools compared: sensitivity, specificity, AUC, and accuracy and supported the results with a ROC curve and Precision-Recall visualizations (Figure 3C and Appendix S7, Datasets EV18-EV19).

4. Page 5, line 13 - the observation of a much larger proportion of SEPs in bacterial genomes immediately raises the concern of "noise" in the detection. The authors are attempted to use experimental validation (which is good), but that does not provide evidence in each case. Maybe the authors can be a bit more specific here about control of false positive/false negative parameters and expectations in RanSEPs.

This is a shared criticism with the first reviewer (see above) and we have made a significant effort to improve the control of false detections. Specifically addressing this comment, we have considerably improved the validation section in results ('RanSEPs validation and method comparative'). First, we increased the size of the positive validation set from 135 to 570 with SEPs detected by MS in 12 different bacterial genomes (new MS experiments have been done on the current version for three bacterial species). For the negative set and using the same species included in the positive set, we took as true negatives a collection of 14,746 putative small open reading frames satisfying: i) not conserved in closely related species, ii) more than 2 high responsive UTPs by PeptideSieve and iii) not detected by MS (Dataset EV19).

Then, we performed two different analyses to assess RanSEPs predictions. First, we performed a control of the predictions with the positive set and a randomly sampled balanced set of negatives (N=570) extracted from the negative set. We explored and compared the predictions with 5 other tools in terms of sensitivity, specificity, AUC and accuracy, and supported the results with a ROC curve and Precision-Recall visualizations (Figure 3C and Appendix EV7, Datasets EV18-EV19). On the other hand, we evaluated the false positive/negative rates simply by using the whole negative set. Glimmer and CPC yielded the lowest FPRs but also had significantly limited TPRs. The rest of the tools presented comparable FPRs, with values of 5.1%, 4.3%, 3.6% and 3.9% for Prodigal, RanSEPs, BASys and GeneMarkS, respectively (Dataset EV19).

5. Page 6, line 8 - the "decoy dataset" is confusing and needs to be clarified a bit in this text section. It is detailed a bit better in the Materials section, but not clear here.

We have added a brief description of the 'decoy' dataset in the results section: the last part of the first paragraph in the first section entitled 'Key factors and criteria for the experimental identification of SEPs':

"A 'decoy' dataset with a comparable database size (20,100 smORFs and 1,608 ORFs), and the same base composition and codon adaptation index (CAI) as *M. pneumoniae*, was used

as a negative control to detect possible MS artifacts (Dataset EV3; see Material and Methods).”

In addition, we have modified the material and methods ‘databases’ sections to better explain the different datasets used. We split the section into two different parts: ‘ORFomes databases generation’ and ‘Decoy databases generation’. For the latter section, we have included a detailed description about how we generated the ‘decoy’ dataset.

6. Page 6, bottom - as the authors point out, the > 2 UTP requirement may be too stringent for these small proteins. Why not consider 1 UTP plus one "other" peptides, whether it is unique or non-unique? This is more consistent with how much of the proteome field uses thresholds. The requirement of >2 UTP is too restrictive. Was mass accuracy enforced on these peptides? That would provide additional restrictions.

We have extensively modified the first section of the results (now entitled ‘Key factors and criteria for the experimental identification of SEPs’) to include observations obtained when adding the thresholds suggested (≥ 1 UTP and ≥ 1 NUTP). By targeted proteomics we concluded that the threshold suggested was not appropriate for defining a set of actual novel SEPs, as it presented false positive putative SEPs (signal by label free proteomics but not targeted MS). These observations are exposed in paragraph 2 of the first section of results. Figure 2 (panels B, C and D) and table 1 have been updated to include these observations.

Additionally, we have extended and detailed the mass spectrometry protocol in the Material and Methods section including information about mass accuracy enforcement (last line in Material and Methods; ‘Mass spectrometry analysis; database search’):

“Searches were performed using a mass accuracy enforcement of 7 ppm, which fits accordingly with the accuracy of the orbitrap mass analyzer, and a product ion tolerance of 0.5 Da. Resulting data files were filtered for $FDR < 1$.”

7. Page 7, bottom - this discussion of RanSEP is nice and somewhat offsets my comment #4 above, but there is still some confusion about how TPR and FPR are set and validated?

The validation section in results (‘RanSEPs validation and method comparative’) has been rewritten to include information about how we define the validation datasets (extended details in Material and Methods), the software comparative in terms of accuracy, TPR and FPR, and a last test defined to control and evaluate the impact of false positives on our predictions.

8. Page 9, bottom - the authors should be careful here in the header to state that "RanSEP is the only tool..." There are other available, and have been demonstrated on other systems, including plants.

The header has been changed to ‘RanSEPs validation and method comparative’ to better match the content described in this section. In addition, we carefully studied the available tools in order to include them in the comparative. However, previous studies on the detection of SEPs did not include a platform/software to perform new predictions, so we decided to include the validated

examples presented in Friedman et al, 2017 and Hücker et al, 2017 (SEPs detection in *H. pylori* and *E. coli*, respectively).

2nd Editorial Decision

18th December 2018

Thank you again for submitting your work to Molecular Systems Biology. We have now heard back from the referees who accepted to evaluate the revised study. As you will see, referee #2 is now positive. Reviewer #1 is however less supportive.

Reviewer #1 was not convinced that the ribosome profiling data provide 'unequivocal' evidence and would have preferred to see confirmation by Western blotting. We agree that having such evidence would be much better and we strongly encourage you to provide these if available. On the other hand, given the potential difficulties mentioned with regard to conducting conclusive validation by Western blot, and to unblock the situation, we would ask that you tone down the strength of the conclusions derived from the ribosomal profiling data and, indeed as suggested by reviewer #1, present RanSEP as a 'prediction tool' rather than an 'annotation tool'. Point #2 raised by reviewer #1 should also be explicitly addressed. With regard to point #3, we would also ask to reduce speculations as suggested by the reviewer.

REFeree REPORTS

Reviewer #1:

The identification of novel small open reading frames is essential for full understanding of the complex layers of regulation that occur within each cell, and computational approaches such as RanSEPs could have an integral role in the process of uncovering these genes. This version of the manuscript is improved, in many ways, over the previous version (the increased stringency for utilization of the mass spectrometry data removes false positives that were present before, and the ribosome profiling data serves as an independent source of data to support those hits) and represents a significant amount of work.

However, some key issues were insufficiently addressed, leaving this reviewer unconvinced of the efficacy of this method for the annotation of new SEPs. The core issue, as mentioned in the previous review, is that the mass annotation of genomes for SEPs without sufficient corroboration could do more harm than good. Thus, it was disappointing that the authors refused to experimentally verify predicted proteins by immunoblot detection of tagged proteins expressed from the chromosome, despite being able to choose from 12 different organisms in which to carry out the validation. Although RanSEPs identifies many known smORFs, the goal is to demonstrate that new genes with detectable products can be identified with this method. One reason given for not carrying out experimental validation (that detection of small proteins by immunoblot analysis has a low success rate) demonstrates a lack of understanding - both of the field and of the study that was cited. The point is that many are false predictions (coming from computational analysis, mass spectrometry and ribosome profiling). Contrary to the authors claim, ribosome profiling data does not "unequivocally confirm the existence of a SEP". While RanSEPs may be an incredibly powerful tool for predicting possible SEPs, it does not meet the burden of proof for small protein annotation. The manuscript would be more acceptable if the authors denoted RanSEPs a predictive tool rather than an annotation tool.

Additional comments:

1. The authors' response mentions a 77% success rate for the proteins that were identified in a recent study (Van Orsdel et al. 2018) as a means of providing credibility of RanSEPs over conventional tagging. What is the success rate for the candidates not detected by western analysis?
2. An obvious route of obtaining supporting data for the predictions was the examination of available ribosome profiling data for the predicted new SEPs of *E. coli* (which was used as validation of the mass spectrometry hits). Was this done or is there a reason this was not carried out?
3. The "functional analysis" of the SEPs carried out by the authors consists primarily of using BLAST and Phobius. While these are both useful tools, many proteins are misannotated or

annotated solely by homology - both of these lead to the propagation of incorrectly annotated functions, especially for sequences that are short and/or are poorly conserved. Additionally, the emphasis that any predicted transmembrane or secreted SEP has a role in quorum sensing or as a toxin ignores a wide range of possibilities associated with poorly conserved small proteins. The presence of "a N-terminus predicted signal peptide" cannot be equated with a role in "quorum sensing and/or signaling". The authors may want to better familiarize themselves with the literature regarding what is known about small proteins with a transmembrane domain in both bacteria and eukaryotes (for example, AcrZ, myoregulin and sarcolipin). In general, the authors should limit their speculation about possible functions.

4. The text was significantly improved by the external review by a native English scientific editor, but grammatical errors persist.

5. More minor comments:

--Introduction: The authors should update their citations for reviews on small proteins as well as primary literature on their functions. Much has been learned in the past 10 years. As just one example, the authors do not cite Van Orsdel et al. 2018 mentioned in their response.

--Expanded view data sets: The authors need to provide a key or table of contents for their Data sets.

Reviewer #2:

Authors have done a careful and thorough job of responding to the detailed reviewer comments. In particular, they added key new work, clarified poor quality text, expanded/clarified validation approaches, and better qualified the novelty and framework of their new tool.

While I have some remaining minor concerns about this overall approach, much of these reflect the general field rather than the authors' specific work. In general, I feel that they have now done a detailed and appropriate job in carefully and defensibly defining their approach. As such, I believe that this manuscript is now suitable for publication consideration.

2nd Revision - authors' response

11th January 2019

Reviewer #1 was not convinced that the ribosome profiling data provide 'unequivocal' evidence and would have preferred to see confirmation by Western blotting. We agree that having such evidence would be much better and we strongly encourage you to provide these if available. On the other hand, given the potential difficulties mentioned with regard to conducting conclusive validation by Western blot, and to unblock the situation, we would ask that you tone down the strength of the conclusions derived from the ribosomal profiling data and, indeed as suggested by reviewer #1, present RanSEP as a 'prediction tool' rather than an 'annotation tool'. Point #2 raised by reviewer #1 should also be explicitly addressed. With regard to point #3, we would also ask to reduce speculations as suggested by the reviewer.

We have carefully ensured that RanSEPs is presented as a prediction tool rather than an annotation tool throughout the manuscript to avoid misleading conclusions regarding the usage of RanSEPs. Changes that reflect this are:

- 'Genome annotation' in the keywords section has been replaced by 'protein prediction'.
- Fourth line, paragraph 4 of the 'Introduction' section: 'we developed RanSEPs, a random forest-based tool for the unbiased identification of SEPs in any bacterial

genome' has been changed to 'we developed RanSEPs, a random forest-based tool for the unbiased prediction of SEPs in any bacterial genome'.

- Fourth line from the end of the last paragraph in the 'Introduction' section: 'thereby suggesting that they could play important roles in quorum sensing or signalling' has been toned down and improved with additional references:

'As previously described (Kemp & Cymer, 2014; Sheng et al, 2017), we observed a significant proportion of SEPs with transmembrane segments (9.7%). At a time when deep sequencing of microbiomes results in the identification of thousands of new bacterial species, our tool opens up the possibility to predict new SEPs that could modulate bacterial populations through quorum sensing or antimicrobial properties (Duval and Cossart 2017).'
- The 'Discussion' section clearly concludes now that RanSEPs is intended to be a tool for computer-aided prediction/prioritization of SEPs rather than a tool for annotation: 'Considering these results, we envision RanSEPs as an approach to predict new SEPs, support detections and discard artifactual proteins detected by MS with only one UTP and/or one NUTP. When no experimental information is available, RanSEPs can guide the selection of potential new SEPs for validation and further characterization to depict their functions.'
- Fourth line, paragraph 3 in 'RanSEPs validation and method comparative': 'RanSEPs was the best tool for SEPs annotation (AUC=0.95; accuracy=0.89)' has been modified to 'RanSEPs was the best tool for SEPs prediction (AUC=0.95; accuracy=0.89)'.

Points #2 and #3 have been addressed in the latest version of the article. We have included the relationship between RanSEPs scores and the *E. coli* ribosome profiling signal as suggested by Reviewer #1 in the validation section of the manuscript. With regards to point #3, we have toned down the conclusions derived from the functional analysis and reduced the speculation to better represent the results obtained.

Finally, as suggested by the reviewer, we have included additional references in the 'Introduction' section of the manuscript and an 'Index of Datasets' in the Appendix supplementary information (as suggested in the additional minor comments provided by Referee #1).

Reviewer #1:

The identification of novel small open reading frames is essential for full understanding of the complex layers of regulation that occur within each cell, and computational approaches such as

RanSEPs could have an integral role in the process of uncovering these genes. This version of the manuscript is improved, in many ways, over the previous version (the increased stringency for utilization of the mass spectrometry data removes false positives that were present before, and the ribosome profiling data serves as an independent source of data to support those hits) and represents a significant amount of work.

We appreciate the reviewer for pointing out the strengths of our approach and we agree that the computational prediction of these proteins is an important element in the field of protein discovery. We have made a great effort to improve the quality of the manuscript and thus appreciate the reviewer for pointing this out.

However, some key issues were insufficiently addressed, leaving this reviewer unconvinced of the efficacy of this method for the annotation of new SEPs. The core issue, as mentioned in the previous review, is that the mass annotation of genomes for SEPs without sufficient corroboration could do more harm than good. Thus, it was disappointing that the authors refused to experimentally verify predicted proteins by immunoblot detection of tagged proteins expressed from the chromosome, despite being able to choose from 12 different organisms in which to carry out the validation. Although RanSEPs identifies many known smORFs, the goal is to demonstrate that new genes with detectable products can be identified with this method. One reason given for not carrying out experimental validation (that detection of small proteins by immunoblot analysis has a low success rate) demonstrates a lack of understanding - both of the field and of the study that was cited. The point is that many are false predictions (coming from computational analysis, mass spectrometry and ribosome profiling). Contrary to the authors claim, ribosome profiling data does not "unequivocally confirm the existence of a SEP". While RanSEPs may be an incredibly powerful tool for predicting possible SEPs, it does not meet the burden of proof for small protein annotation. The manuscript would be more acceptable if the authors denoted RanSEPs a predictive tool rather than an annotation tool.

We understand the concern of the reviewer and would like take advantage of this revision to further clarify our approach and the assertions made throughout the study.

First of all, we agree with the reviewer's concern that 'mass annotation of genomes for SEPs without sufficient corroboration could do more harm than good'. In the first round of revision we understood that the reviewer's concern was related to using ≥ 2 UTPs as the benchmark for validating our predictions. That is why we supported such candidates by integrating ribosome profiling and demonstrating that SEPs detected with ≥ 2 UTPs also present higher coverages by ribosome profiling (Appendix Figure S5).

Using the study of Van Orsdel *et al.* 2018 as a reference, we found that RanSEPs positively predicted 29 out of the 36 SEPs validated by immunoblotting but only 3 of the 44 putative smORFs not validated by immunoblotting. This could be interpreted as RanSEPs having a success rate of 80% with a false positive rate of 6.8%. However, one of the SEPs not detected by immunoblotting but predicted as positive by RanSEPs (pntA_ydgH_2 in the reference study) satisfied the ≥ 2 UTPs

criteria and presented positive features such as the presence of a ribosome binding site and a signal in ribosome profiling. The other 2 SEPs predicted as positive by RanSEPs that were not detected by immunoblotting presented the same coding features. We have no means to conclude whether these proteins are false negatives with regards to immunoblotting or false positives with regards to mass spectrometry, ribosome profiling and RanSEPs. Thus, we believe that although mass spectrometry is not free from false positives, detection via immunoblotting can also present limitations and possibly misses candidates that are expressed only under a specific condition and candidates that have short half-lives. Considering the high level of agreement between mass spectrometry and Western blot for 36 SEPs, we assumed that mass spectrometry (now supported by ribosome profiling) was a valid additional approach to provide a genome-wide perspective. The 80 validated cases by Van Orsdel *et al.* 2018 (36 positive cases and 44 negatives) were originally included in the validation set used in the validation section of our manuscript, and they contributed to the prediction quality metrics presented in the ‘RanSEPs validation and method comparative’ section.

We understand the problem behind the indiscriminate annotation of SEPs and that is what motivated us to carefully estimate the number of false positives using different approaches in the ‘RanSEPs validation and method comparative’ section. We demonstrated that RanSEPs predicts SEPs with greater accuracy (0.89) than other computational tools with a comparable false positive rate. This accuracy is not only based on SEPs detected by mass spectrometry but includes additional candidates identified by targeted proteomics and immunoblotting from different bacterial genomes. We understand and agree that we cannot consider RanSEPs as an annotation tool but rather as a prediction tool, and we apologize if it was not correctly expressed throughout the manuscript. Thus, we have now carefully ensured that RanSEPs is presented as a prediction tool and not as an annotation tool to avoid misleading conclusions regarding the usage of RanSEPs. Changes that reflect this are:

- ‘Genome annotation’ in the keywords section has been replaced by ‘protein prediction’.
- Fourth line, paragraph 4 of the ‘Introduction’ section: ‘we developed RanSEPs, a random forest-based tool for the unbiased identification of SEPs in any bacterial genome’ has been changed to ‘we developed RanSEPs, a random forest-based tool for the prediction of SEPs in any bacterial genome’.
- Fourth line, 3rd paragraph in ‘RanSEPs validation and method comparative’: ‘RanSEPs was the best tool for SEPs annotation (AUC=0.95; accuracy=0.89)’ modified to ‘RanSEPs was the best tool for SEPs prediction (AUC=0.95; accuracy=0.89)’.

In any case, we believe that an important application of RanSEPs could be to prioritize the candidates which should be included in future mass spectrometry searches and to guide which candidates should be further explored by immunoblotting or other techniques. This idea has now been included in the discussion of the manuscript as:

‘When no experimental information is available, RanSEPs can guide the selection of candidate SEPs for validation and further characterization.’

Additional comments:

1. The authors' response mentions a 77% success rate for the proteins that were identified in a recent study (Van Orsdel *et al.* 2018) as a means of providing credibility of RanSEPs over conventional tagging. What is the success rate for the candidates not detected by western analysis?

Following our previous comment, we believe that this has been misunderstood. In our reference study Van Orsdel *et al.* 2018, the authors presented 80 smORFs as real based on strong evidence: they did not overlap with other known annotations and they had a significant ribosome profiling coverage matching the annotation. Out of this group, 36 were validated by immunoblot analysis. We validated these with our experimental mass spectrometry data and found that 28 of the 36 SEPs (77%) satisfied our ≥ 2 UTPs criteria, thereby reinforcing the reliability of this criteria when used to select positive candidates based on mass spectrometry. We were not referring to the success rate of RanSEPs but rather to the success rate of detection provided by mass spectrometry. This can be seen in the first round of revision:

In reference to the question formulated by the reviewer, below we include the relationship between mass spectrometry (MS), Western blot (WB) and RanSEPs (RS) predictions. In the table, RS+ indicates RanSEPs predicted a score ≥ 0.5 , and MS+ indicates that the candidate appeared with ≥ 2 UTPs in mass spectrometry searches.

	Total cases	RS+	RS-
WB + MS +	28	24	3
WB + MS -	8	5	3
WB - MS +	2	1	1
WB - MS -	42	2	40
TOTAL	80	33	47

RanSEPs correctly predicted 29 cases out of the 36 validated by Van Orsdel *et al.* 2018, with 24 of them presenting signals in both MS and WB. Out of the 6 cases that were predicted as negatives, 3 did not appear in MS.

The 80 validated cases by Van Orsdel *et al.* 2018 (36 positive and 44 negative cases) were originally included in our validation set (Dataset EV18) and contributed to the prediction quality metrics presented in the 'RanSEPs validation and method comparative' section.

2. An obvious route of obtaining supporting data for the predictions was the examination of available ribosome profiling data for the predicted new SEPs of *E. coli* (which was used as

validation of the mass spectrometry hits). Was this done or is there a reason this was not carried out?

As described in our previous comment, we used ribosome profiling data to further support using the ≥ 2 UTPs criteria when selecting positive candidates (Appendix supplementary Figure S5).

We have taken into account the recommendations provided by the editor and extended the validation section in the manuscript to include the relationship between positively predicted SEPs in *E. coli* by RanSEPs and the ribosome profile coverage represented as RCV (the ratio of transcripts with bound ribosomes, normalized by general expression). With this analysis, we observed that positively predicted SEPs have higher RCV ratios and are more similar to annotated genes than those predicted to be negative and resembling annotated ncRNAs. This is now included in the manuscript in the fourth paragraph of 'RanSEPs validation and method comparative' and as the main result of panel C in Figure 3:

'Finally, we further validated our prediction tool at the genome-wide level by studying the correlation between gene-expression corrected Ribo-Seq coverage (RCV) and RanSEPs' prediction in *E. coli* (Hücker et al, 2017). We found that SEPs predicted as positive showed significantly higher RCV levels compared with candidates predicted as negatives (Mann-Whitney one-sided test p-value= 1×10^{-7}) and ncRNAs (Mann-Whitney one-sided test p-value= 1×10^{-4} , Figure 3C). Additionally, while RanSEPs positive predictions presented RCV values closer to the scores of annotated proteins, although still significantly lower (Mann-Whitney two-sided test, p-value= 1×10^{-10}), negative predictions were more similar to annotated ncRNAs (no significant differences by Mann-Whitney two-sided test, p-value=0.13).'

Although we agree with the reviewer that ribosome profiling presents false positives, we believe that overall it still supports RanSEPs predictions.

3. The "functional analysis" of the SEPs carried out by the authors consists primarily of using BLAST and Phobius. While these are both useful tools, many proteins are misannotated or annotated solely by homology - both of these lead to the propagation of incorrectly annotated functions, especially for sequences that are short and/or are poorly conserved. Additionally, the emphasis that any predicted transmembrane or secreted SEP has a role in quorum sensing or as a toxin ignores a wide range of possibilities associated with poorly conserved small proteins. The presence of "a N-terminus predicted signal peptide" cannot be equated with a role in "quorum sensing and/or signaling". The authors may want to better familiarize themselves with the literature regarding what is known about small proteins with a transmembrane domain in both bacteria and eukaryotes (for example, AcrZ, myoregulin and sarcolipin). In general, the authors should limit their speculation about possible functions.

The problem of missannotations is already considered in the manuscript:

‘Noteworthy, genome annotations are critical for classifying a SEP as a new protein. In fact, for 76% of the SEPs predicted by RanSEPs, orthologous SEPs were identified by Blast in closely related strains. This result indicates that reference genomes are still incomplete and not curated.’

To provide a more cautious idea about the function estimation, we mentioned the concerns at the end of the second paragraph of the ‘Functional assessment of novel SEPs’ section in the results:

‘Although we have assigned functionality to most of the predicted SEPs in the 109 genomes, one needs to be cautious as sequence homology and functional annotation of small proteins is not always reliable.’

We have also toned down the conclusions made throughout the discussion in order to provide a more cautious point of view regarding the functions and the relationship with signalling and quorum sensing. This can be seen in the previous to last paragraph in the discussion section:

‘However, this analysis should be taken with caution as sequence homology and functional annotation of SEPs is challenging (VanOrsdel et al, 2018).’

We agree with the reviewer about our conclusions regarding the quorum sensing and signalling for the predicted SEPs and we have modified the manuscript to mention only the enrichment in polypeptides with at least one membrane segment, supporting the idea with two additional references. This change has been applied to the last paragraph in ‘Introduction’ section:

‘As previously described (Kemp & Cymer, 2014; Sheng et al, 2017), we observed a significant proportion of SEPs with N-terminus predicted signal peptide (9.7%) and transmembrane segments (15%).’

And in the ‘Discussion’ section:

‘Interestingly, similar to what has been previously reported (Kemp & Cymer, 2014; Sheng et al, 2017), we found a significant enrichment in SEPs presenting features indicative of being secreted (10%) or membrane localised (15%). This observation could have an impact not only on translational research but also on the study of the modulation of bacterial populations in microbiomes, thereby opening up a new line of research in the Systems Biology discipline (Duval and Cossart 2017).’

4. The text was significantly improved by the external review by a native English scientific editor, but grammatical errors persist.

We apologize for the grammatical errors, they could have occurred while incorporating the revisions during the previous round. We have now carefully reviewed the text to double check and correct any remaining grammatical errors. Furthermore, previous to this latest submission, the manuscript and all the additional content has been reviewed again by a native English scientific editor.

5. *More minor comments:*

--Introduction: The authors should update their citations for reviews on small proteins as well as primary literature on their functions. Much has been learned in the past 10 years. As just one example, the authors do not cite Van Orsdel et al. 2018 mentioned in their response.

We apologize for the lack of citations and have now included Van Orsdel et al 2018 in the text. Other citations added to the latest version of the manuscript are:

- We now provide a more detailed list of functional examples and potential application in the first paragraph of the introduction with the relevant references:

‘In bacteria, SEPs exhibit a wide range of functions that are essential for the cell. SEPs can be involved in cell division (Blr, MciZ and SidA), transport (AcrZ, KdpF and SgrT), signal transduction (MgrB and Sda) or even act as chaperones (FbpB, FbpC and MntS) (Storz et al. 2014). They are also involved in protein complexes, stress responses, virulence, and sporulation (Burkholder et al, 2001; Rowland et al, 2004; Alix & Blanc-Potard, 2008; Hemm et al, 2010; Lluch-Senar et al, 2015). Interestingly, these small proteins can also be used for communication between bacteria and/or phages, and as bacteriocins within niches like microbiota, thereby making them an important molecule to study when searching for new therapeutic protein candidates (Duval and Cossart 2017).’
- Van Orsdel et al 2018 is now cited in the last paragraph of ‘Introduction’:

‘This result suggests that a remarkable number of bacterial SEPs remains unexplored, as recently reported (VanOrsdel et al, 2018).’
- To support our results regarding the significant number of SEPs with transmembrane segments studies by Kemp & Cymer (2014) and Sheng (2017) have been included as mentioned above in the point 3.

--Expanded view data sets: The authors need to provide a key or table of contents for their Data sets.

We followed the indications in the ‘Author Guidelines’ section of the journal webpage by including a first sheet with the legend of the dataset for each expanded dataset. In order to make the exploration of these datasets easier, we have now included an ‘Index of datasets’ in the Supplementary information (see Appendix). This has been indicated in the ‘Data and Software availability’ section.

Reviewer #2:

Authors have done a careful and thorough job of responding to the detailed reviewer comments. In particular, they added key new work, clarified poor quality text, expanded/clarified validation approaches, and better qualified the novelty and framework of their new tool.

While I have some remaining minor concerns about this overall approach, much of these reflect the general field rather than the authors' specific work. In general, I feel that they have now done a detailed and appropriate job in carefully and defensibly defining their approach. As such, I believe that this manuscript is now suitable for publication consideration.

We appreciate that the reviewer has pointed out the improvements made during the revision process. We also agree that there are still problems and missing knowledge in the field of small protein characterization that should be addressed in future work. Nevertheless, we believe that RanSEPs can help to overcome such limitations and we appreciate the opportunity to publish this work and make it available to the scientific community.

3rd Editorial Decision

21st January 2019

Thank you again for sending us your revised manuscript. We are now satisfied with the modifications made and I am pleased to inform you that your paper has been accepted for publication.

Corresponding Author Name: **Maria Lluch-Senar**Journal Submitted to: **Molecular Systems Biology**Manuscript Number: **MSB-18-8290R****